# ACRL: Adaptive Control of Training-Inference Discrepancy for Stable Reinforcement Learning

## Abstract

Reinforcement Learning (RL) training for Large Language Models (LLMs) often suffers from instability due to the discrepancy between training and inference. This training-inference discrepancy stems from two primary factors: an architectural separation between training and inference engines, and the use of low-precision quantization in inference versus higher-precision computation in training. To address training instability issues caused by high training-inference discrepancy, we present the principles and methods for its adaptive control. We propose **Adaptive Control Reinforcement Learning (ACRL)**, which adaptively maintains the training-inference discrepancy within a reasonable range to ensure stable RL training. Beyond stabilization, ACRL inherently increases policy entropy, thereby enhancing exploration and improving accuracy. The experimental results show that when the inference engine utilizes FP8 quantization, ACRL consistently maintains the training-inference discrepancy within a reasonable range and stabilizes RL training. Furthermore, ACRL not only matches the accuracy of the BF16 baseline but also outperforms importance sampling (IS) fixes.

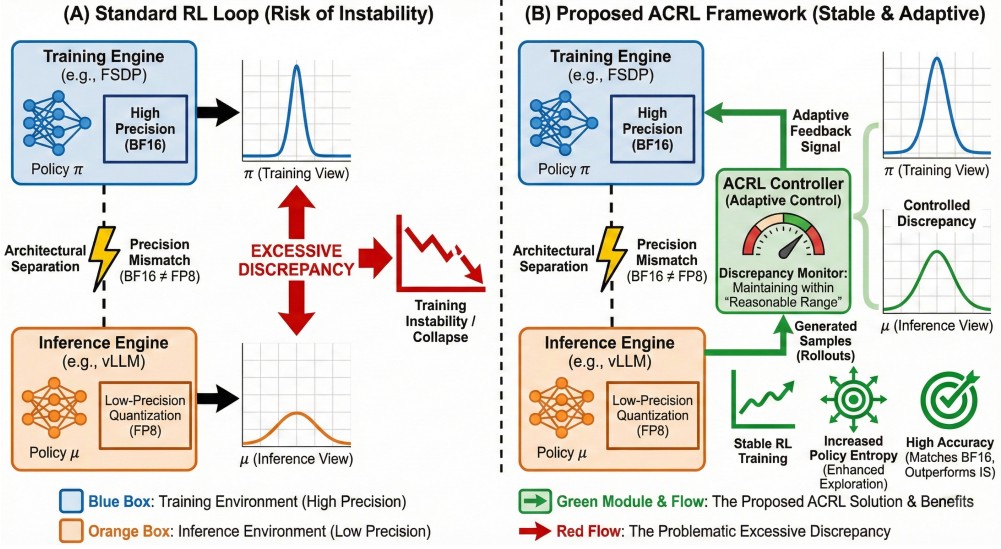

Figure 1: **Overview of the ACRL framework.** (A) The standard RL training loop suffers from instability and potential collapse due to an excessive discrepancy between the high-precision training policy ($\pi$) and the low-precision, quantized inference policy ($\mu$). (B) The proposed Adaptive Control Reinforcement Learning (ACRL) framework introduces a controller that adaptively maintains this training-inference discrepancy within a reasonable range. This active regulation provides a stable adaptive feedback signal, ensuring robust RL training, inherently increasing policy entropy for enhanced exploration, and achieving accuracy that matches BF16 baselines while outperforming standard importance sampling (IS).

# 1 Introduction

Reinforcement learning has drawn immense attention in LLM training since InstructGPT Ouyang et al. (2022) first demonstrated the significance and benefits that reinforcement learning can bring to LLM training. Many LLMs, including DeepSeek Shao et al. (2024), Qwen Yang et al. (2025), and Gemini Kamath et al. (2025), trained by reinforcement learning algorithms, come out every year. Popular frameworks, such as VeRL Sheng et al. (2025), which incorporates various reinforcement learning algorithms such as PPO Schulman et al. (2017) and GRPO Shao et al. (2024), significantly reduce the engineering difficulty of using RL in LLM post-training.

Modern large language models are being developed with increasingly large parameter sizes that introduce significant computational overhead and memory consumption. Given that inference/rollout accounts for the majority of total learning time, enhancing inference efficiency has become imperative. Therefore, current practices utilize dedicated high-performance inference engines (e.g, vLLM and SGLang), which are separated from training engines (e.g, FSDP and Megatron). Inference engines apply low-precision quantization, e.g, FP8 Kuzmin et al. (2022), MXFP8 Mishra et al. (2025), INT8 Dettmers et al. (2022); Xiao et al. (2023), to accelerate computations and save memory consumptions.

The use of efficient inference engines and low-precision quantization significantly accelerate RL training. However, operation kernels, low-precision quantization, and other factors in those efficient inference engines lead to a mismatch in probability distributions between training and inference Yao et al. (2025); Qi et al. (2025), which we call the training-inference discrepancy. The discrepancy propagates through the policy, turning on-policy learning into off-policy learning, significantly amplifying instability during learning and leading to training collapse Liu et al. (2025a). Current solutions aim to resolve the mismatch using algorithmic patches or precision agreement/alignment. A token-level IS patch Yao et al. (2025) on GRPO attempts to reduce the impact of the training-inference discrepancy by applying a ratio to the gradient on the token level without considering the sequence-level discrepancy. Subsequently, sequence-level IS Liu et al. (2025a) applies a single importance ratio over the entire generated sequence, but it results in vanishingly small ratios and fails to converge on low-precision data types. On the other hand, previous literature LMSYS Org (2025); Qi et al. (2025) makes alignment on the precision used in both inference and training. Applying FP8 LMSYS Org (2025) uniformly reduces the training-inference discrepancy compared to BF16 and stabilizes the training, but it results in a loss of accuracy. Moreover, using FP16 Qi et al. (2025), which is in higher-precision and has more representation power than BF16, gives higher accuracy, but such alignment in FP16 does not fully leverage the computational efficiency of low-bit quantization. Critically, the alignment-based methods cannot be generalized to fix other discrepancies He & Lab (2025) between engines (e.g, operation kernels) or be applied to different inference algorithms Draye et al. (2025).

Therefore, we propose Adaptive Control Reinforcement Learning (ACRL) to adaptively control the discrepancy between training and inference near a reference value to stabilize the training (see overview in Figure 1). ACRL uses measurements of the sequence-level discrepancy as a reference to dynamically adjust the token-level gradient ratio. It prevents the discrepancy from amplifying into training collapse and prevents the discrepancy from diminishing, which leads to a loss in accuracy. Moreover, ACRL encourages more exploration and produces models that achieve higher overall accuracy. Empirically, we show that ACRL effectively controls the inference-training discrepancy under aggressive FP8 quantization, ensuring stable RL training. On mathematical reasoning benchmarks (GSM8K, AIME, HMMT, AMC, MATH500), ACRL-trained models achieve comparable accuracy to the BF16 baseline and show substantial improvements over previous IS fixes.

In summary, this paper makes the following contributions:

- We propose **a new perspective for stable RL** training by controlling the training-inference discrepancy within a reasonable range and elucidate its underlying principles.

- We propose ACRL, an algorithm that applies token-level gradient ratios based on sequence-level measurements that adaptively control the training-inference discrepancy. ACRL **ensures stable RL training and inherently enhances exploration to improve the accuracy**.

- Our empirical validation demonstrates that **ACRL enables stable RL training under aggressive FP8 quantization schemes, with the accuracy not only matching the BF16 baseline but also outperforming IS fixes**.

## 2 Preliminaries

In modern RL frameworks for LLM training, e.g, VeRL Sheng et al. (2025), different engines are used for inference and training to maximize system efficiency, which inevitably creates a mismatch between the inference policy and the training policy. Let $\mu$ denote the inference policy and $\pi$ the training policy. In principle, $\mu$ and $\pi$ should be identical due to the on-policy nature of the training. In reality, $\mu \neq \pi$ due to precision loss in quantization and other implementation differences (e.g, operation kernels).

The popular VeRL framework implements GRPO Shao et al. (2024), DAPO Yu et al. (2025) and some other research variant training algorithms  Li et al. (2024); Zheng et al. (2025); Chen et al. (2025). The standard GRPO objective shown in equation 1 optimizes the policy $\pi$. For every prompt $q$, it samples a set of $G$ responses $\{a_i\}_{i=1}^{G}$ from the inference policy $\mu(\cdot|q,\theta_{old})$ to compute the advantage values $A$.

$$\nabla \mathcal{J}_{grpo}(\theta) = \mathbb{E}_{a_i \sim \mu(\cdot|q,\theta_{old})} \left[ \frac{1}{G} \sum_{i=1}^{G} \frac{1}{|a_i|} \sum_{t=1}^{|a_i|} \nabla_\theta \min \left( r_{i,t} A_{i,t}, \mathrm{clip}(r_{i,t}, 1-\epsilon, 1+\epsilon) A_{i,t} \right) \right] \quad (1)$$

where $r_{i,t} = \frac{\pi(a_{i,t}|q,a_{i,<t},\theta)}{\pi(a_{i,t}|q,a_{i,<t},\theta_{old})}$ and $A_{i,t} = \frac{R_i - \mathrm{mean}(\{R_i\}_{i=1}^{G})}{\mathrm{std}(\{R_i\}_{i=1}^{G})}$.

However, the standard GRPO objective does not account for the policy mismatch between training and inference. While the expectation is computed over samples from the inference policy $\mu$, the importance ratio $r_{i,t}$ is derived using the training policy $\pi$. This inconsistency results in a training-inference discrepancy that effectively turns the policy training into off-policy and can lead to training collapse Liu et al. (2025a).

### 2.1 Quantization

Quantization is an optimization technique that represents data in a lower bit format to save memory consumption and often increases operation speed because modern hardware supports operation kernels on lower-precision data types. However, quantization reduces the numerical precision of model parameters and activations from 32-bit floating-point (FP32) to lower-bit formats (e.g., FP16, INT8). In the example of GRPO, an inference policy that utilizes W8A8 or W4A4 would return a very dissimilar probability distribution compared to the training policy that uses BF16. In practice, the major cause of the training-inference discrepancy is the usage of quantization in the inference engines while training engines use the standard BF16 format.

### 2.2 Importance Sampling

Previous work LMSYS Org (2025); Qi et al. (2025) has tried to agree/align the data format in both the inference engine and the training engine to reduce the training-inference discrepancy. However, this alignment approach does not scale well, because the inference is the bottleneck of the entire training process, and often dominates the computation; if we use FP16/BF16 on both sides, we do not fully leverage hardware optimizations for lower precision. Aggressive quantization (e.g, FP8) could reduce compute and memory consumption, but the precision loss in the training would slow the convergence if not carefully managed.

In particular, IS has been studied to solve the training-inference discrepancy. IS adjusts the gradient calculation using a probability distribution ratio, ensuring the gradient estimator remains unbiased.

Based on GRPO, a token-level truncated importance sampling(TIS) correction Yao et al. (2025); Liu et al. (2025b) has been developed which is shown as:

$$\nabla \mathcal{J}_{grpo-token}(\theta) = \mathbb{E}_{a_i \sim \mu(\cdot|q,\theta_{old})} \left[ \frac{1}{G} \sum_{i=1}^{G} \frac{1}{|a_i|} \sum_{t=1}^{|a_i|} \min(\rho_{i,t}, C) \right.$$
$$\left. \cdot \nabla_\theta \min \left( r_{i,t} A_{i,t}, \text{clip}(r_{i,t}, 1 - \epsilon, 1 + \epsilon) A_{i,t} \right) \right], \tag{2}$$

where $\rho_{i,t} = \frac{\pi(a_{i,t}|q,a_{i,<t},\theta_{old})}{\mu(a_{i,t}|q,a_{i,<t},\theta_{old})}$. However, Liu et al. (2025a) suspects the token-level TIS yields a biased gradient estimator. Thus, a sequence-level masked importance sampling(MIS) variant Liu et al. (2025a) shown in equation 3 has been developed with an unbiased gradient estimator. The correction is applied to the entire gradient term, using a single ratio for the whole sequence, which has large variances, producing vanishingly small importance weights that slow convergence.

$$\nabla \mathcal{J}_{grpo-sequence}(\theta) = \mathbb{E}_{a_i \sim \mu(\cdot|q,\theta_{old})} \left[ \frac{1}{G} \sum_{i=1}^{G} \frac{1}{|a_i|} \rho_i \cdot \mathbb{I}\{\rho_i \le C\} \right.$$
$$\left. \sum_{t=1}^{|a_i|} \nabla_\theta \min \left( r_{i,t} A_{i,t}, \text{clip}(r_{i,t}, 1 - \epsilon, 1 + \epsilon) A_{i,t} \right) \right], \tag{3}$$

where $\rho_i = \frac{\pi(a_i|q,\theta_{old})}{\mu(a_i|q,\theta_{old})}$.

## 3 Training-Inference Discrepancy Control

This section establishes the motivations for and mechanisms of training-inference discrepancy control.

### 3.1 Motivations for Regulating Training-Inference Discrepancy

**To prevent training collapse, the training-inference discrepancy should not be excessively large.**
If the training-inference discrepancy is excessively large, the inference policy $\mu$ and the training policy $\pi$ differ significantly. During policy updates, the training policy $\pi$ is adjusted using the samples generated by $\mu$, which introduces bias (or distribution mismatch error) into the policy gradient. Over time, these erroneous updates accumulate and destabilize the training to collapse Liu et al. (2025a).

**To avoid accuracy loss, the training-inference discrepancy should not be excessively small.** The training-inference discrepancy $E$ can be defined as

$$E = f(Q, \theta, \eta) \ge 0, \tag{4}$$

where $Q$ denotes the input prompts, $\theta$ represents the model parameters, and $\eta$ is a random error. The function $f(\cdot)$ is a complex function. It is governed by the following intrinsic factors: (a) Discrepancies in numerical precision, such as the application of low-bit quantization during inference versus higher precision in training. (b) Inconsistent backend implementations, where training and inference engines (e.g., FSDP and vLLM) may use different low-level implementations. (c) Noise stemming from runtime-dependent nondeterminism, including dynamic batching, non-deterministic kernel execution, parallel reduction order, etc. When there is no discrepancy, i.e., $E = 0$, the training and inference policies assign identical probabilities to all evaluated tokens.

For a fixed dataset and runtime configuration, we can only update the model parameters $\theta$ to systematically adjust the discrepancy $E$. Let $\mathcal{J}(\theta)$ denote the original reinforcement-learning objective. Without an explicit discrepancy constraint, the optimization problem is

$$\max_\theta \quad \mathcal{J}(\theta). \tag{5}$$

If the training policy is additionally required to remain within a discrepancy bound $E_0$, the optimization problem becomes

$$\begin{cases} \max\limits_{\theta} & \mathcal{J}(\theta) \\ \text{s.t.} & E \leq E_0. \end{cases} \tag{6}$$

This constraints restricts the feasible parameter space to

$$\Theta_{E_0} = \{\theta \,:\, E \leq E_0\}. \tag{7}$$

For any $E_1 < E_2$, the selectable parameter region $\Theta_{E_1} \subseteq \Theta_{E_2}$. Therefore, forcing the discrepancy to be excessively small restricts the parameter space. If an optimal parameter $\theta^*$ has $E^* > E_0$, the training would forgo this particular optimal parameter due to the tight discrepancy tolerance which may lead to a decline in training accuracy.

More intuitively, imposing heavy penalties for the discrepancy in the reward function leads to a multi-objective optimization conflict. This forces the model to forgo high-reward weights to reduce the discrepancy. Furthermore, an overly constrained discrepancy encourages the training policy to overfit to the inference policy, which weakens generalization and causes a decline in accuracy.

Therefore, the training-inference discrepancy should be controlled within a reasonable range by reducing it when excessively large and amplifying it when excessively small.

### 3.2 Mechanisms for Regulating Training-Inference Discrepancy

**To prevent the training-inference discrepancy from being excessively large**, we adopt the training probability adjustment principle shown in Figure 2. If the training probability exceeds the inference probability, the training probability should be reduced toward the inference probability. Conversely, the training probability should be increased toward the inference probability. In both cases, the training probability moves toward the inference probability to bridge the training-inference discrepancy.

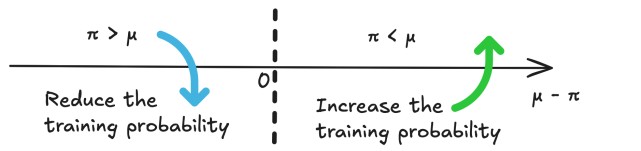

Figure 2: Training Probability Adjustment for Decreasing Training-Inference Discrepancy

Figure 3: AGPA for Decreasing Training-Inference Discrepancy

In the training policy updating, the advantage $A_t$ decides if the action is rewarded or punished. If the advantage is positive, the action is rewarded by increasing its probability. Otherwise, the action is punished by decreasing its probability. We use the principle from Figure 2 to enhance/reduce the magnitude of the reward/punishment. Consequently, we have a four-quadrant policy update rule, which we refer to as the Advantage Guided Probability Adjustment(AGPA) for decreasing training-inference discrepancy, shown in Figure 3 and formalized below:

1. Quadrant I ($A_t > 0$, $\pi < \mu$): The action is beneficial ($A_t > 0$), and $\pi$ is below $\mu$. Both the advantage sign and the update principle agree on increasing $\pi$, so we apply an enhanced positive update.

2. Quadrant II ($A_t > 0$, $\pi > \mu$): The action is beneficial ($A_t > 0$), but $\pi$ is above $\mu$. To avoid the discrepancy becoming too large, we apply a reduced positive update.

3. Quadrant III ($A_t < 0$, $\pi > \mu$): The action is undesirable ($A_t < 0$), and $\pi$ is above $\mu$. Both the advantage sign and the update principle agree on decreasing $\pi$, so we apply an enhanced negative update.

4. Quadrant IV ($A_t < 0$, $\pi < \mu$): The action is undesirable ($A_t < 0$), but $\pi$ is below $\mu$. To avoid the discrepancy becoming too large, we apply a reduced negative update.

This four-quadrant policy update rule implements the principle in Figure 2 to prevent training-inference discrepancy from becoming excessively large while preserving the update direction(reward/punishment) with respect to the advantage sign.

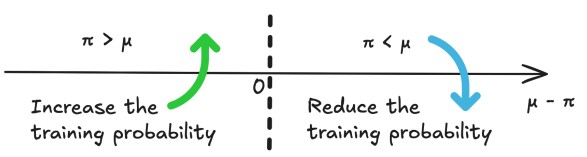 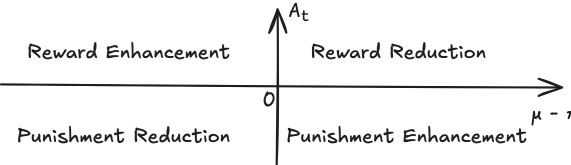

Figure 4: Training Probability Adjustment for Increasing Training-Inference Discrepancy

Figure 5: AGPA for Increasing Training-Inference Discrepancy

**To prevent the training-inference discrepancy from being excessively small**, we apply the complementary, reversed principle illustrated in Figure 4 for training probability update. If the training probability exceeds the inference probability, the training probability should be increased further away from the inference probability. Conversely, the training probability should be decreased further away from the inference probability. In both cases, the training probability moves away from the inference probability to restore the training-inference discrepancy toward the reference value.

This four-quadrant update rule for increasing the training-inference discrepancy, shown in Figure 5, is implemented by adopting the training probability adjustment principle for increasing the training-inference discrepancy shown in Figure 4. We leave its detailed derivation to the reader.

## 4 Algorithm

This section proposes the **Adaptive Control Reinforcement Learning (ACRL)** algorithm to regulate the training-inference discrepancy and improve accuracy.

### 4.1 ACRL Algorithm

We define the distance $d(a_{i,t}, \theta_{old})$ between the training and inference probabilities as follows:

$$d(a_{i,t}, \theta_{old}) \triangleq |\pi(a_{i,t}|q, a_{i,<t}, \theta_{old}) - \mu(a_{i,t}|q, a_{i,<t}, \theta_{old})| \tag{8}$$

*Remark* 1. *This paper employs the absolute distance as the discrepancy metric. Other measures (e.g., Euclidean distance, KL divergence) could be explored in future work.*

We also define a reference value X for the training-inference discrepancy as the average distance over a baseline dataset or initial training step:

$$X \triangleq \frac{1}{n} \sum_{j=1}^{n} \frac{1}{G} \sum_{i=1}^{G} \frac{1}{|a_i|} \sum_{t=1}^{|a_i|} d(a_{i,t}, \theta_{old}) \tag{9}$$

where $n$ is the number of questions. This reference value $X$ is compiled before the training process starts. Computing the reference value $X$ at the beginning offers a key advantage. As seen in equation 4, the model weights $W$ are a primary factor determining the training-inference discrepancy. By using the model's initial weights, which are unaffected by the training-inference discrepancy, we attain a clean baseline measurement of $X$.

To achieve adaptive discrepancy control, we define the ACRL as the following:

$$\nabla \mathcal{J}_{ACRL}(\theta) = \mathbb{E}_{a_i \sim \mu(\cdot|q, \theta_{old})} \left[ \frac{1}{G} \sum_{i=1}^{G} \frac{1}{|a_i|} \sum_{t=1}^{|a_i|} \min(\rho_{i,t}, C) \right.$$

$$\left. \cdot \nabla_\theta \min \left( r_{i,t} A_{i,t}, \text{clip}(r_{i,t}, 1 - \epsilon, 1 + \epsilon) A_{i,t} \right) \right], \tag{10}$$

where

$$\rho_{i,t} = \left( \frac{\pi(a_{i,t}|q, a_{i,<t}, \theta_{old})}{\mu(a_{i,t}|q, a_{i,<t}, \theta_{old})} \right)^\alpha, \tag{11}$$

and

$$\alpha = \gamma \cdot \text{sign}(A_{i,t})(1 - Y/X). \tag{12}$$

Here, $\gamma > 0$ and $C > 1$ are hyper-parameters and $Y = \frac{1}{G} \sum_{i=1}^{G} \frac{1}{|a_i|} \sum_{t=1}^{|a_i|} d(a_{i,t}, \theta_{old})$ is the current sequence's training-inference discrepancy.

ACRL adopts an exponential parameter $\alpha$ to balance the training-inference discrepancy. The parameter $\alpha$ serves two functions. First, it determines the direction of the adaptive update (enhancement or reduction relative to the advantage signal) based on a comparison between Y and X. Second, it incorporates adaptive magnitude into enhancement/reduction. The update strength is positively correlated to the difference between $X$ and $Y$: a larger difference results in a stronger correction, while a smaller difference leads to a gentler correction. Together, this direction and magnitude control constitute the adaptive core of ACRL.

Furthermore, ACRL employs a token-level ratio $\rho_{i,t}$ directly to adjust the update policy. Unlike the sequence-level fix Liu et al. (2025a), which applies a single sequence-wide scaling factor, ACRL uses the sequence-level discrepancy $Y$ to derive token-specific ratios. For each token, $\rho_{i,t}$ is scaled according to the token's individual discrepancy $d(a_{i,t}, \theta_{old})$. Tokens with a large $d(a_{i,t}, \theta_{old})$ receive a stronger adjustment, while tokens with a smaller $d(a_{i,t}, \theta_{old})$ receive a smaller adjustment. This adaptive adjustment mechanism ensures the corrections are applied at local discrepancies, thereby controlling the overall training-inference discrepancy and preserving training stability.

ACRL implements the training-inference discrepancy control principle from Subsection 3.2. Its detailed operational logic is presented in Subsection 4.2.

*Remark 2. The functional form of our adaptive weight, $\rho_{i,t} = (\pi(a_{i,t}|q, a_{i,<t}, \theta_{old})/\mu(a_{i,t}|q, a_{i,<t}, \theta_{old}))^\alpha$, is an empirical engineering solution designed to address practical system failures under aggressive quantization. We acknowledge that this adaptive exponent renders ACRL a biased estimator. However, in modern RL post-training under low-precision constraints, this trade-off is a pragmatic necessity. Theoretically unbiased approaches, such as Sequence-level Masked Importance Sampling (MIS), often suffer from vanishing gradients and severe instability in large-scale practice. The trade-off between theoretical unbiasedness and empirical stability is foundational to modern RL, following the precedence of biased clipping mechanisms in PPO and GRPO. ACRL deliberately prioritizes active discrepancy control over theoretical unbiasedness to ensure optimization stability and recover higher reasoning accuracy in low-precision environments.*

### 4.2 ACRL Mechanism

In ACRL, the ratio $\rho_{i,t}$ is used to directly adjust the policy update. A value of $\rho_{i,t} > 1$ enhances rewards or punishments, while a value of $\rho_{i,t} < 1$ reduces them. ACRL adaptively controls the training-inference discrepancy based on the comparison between the sequence-level training-inference discrepancy $Y$ and the reference discrepancy $X$. If $Y > X$, which suggests that the current sequence-level discrepancy exceeds the reference value, then we decrease the training-inference discrepancy. The adjustments for the four quadrants are as follows (corresponding to Figure 3):

1. Quadrant I : $\alpha < 0$ and $\frac{\pi(a_{i,t}|q, a_{i,<t}, \theta_{old})}{\mu(a_{i,t}|q, a_{i,<t}, \theta_{old})} < 1$. Thus, $\rho_{i,t} > 1$ means reward enhancement.

2. Quadrant II: $\alpha < 0$ and $\frac{\pi(a_{i,t}|q,a_{i,<t},\theta_{old})}{\mu(a_{i,t}|q,a_{i,<t},\theta_{old})} > 1$. Thus, $\rho_{i,t} < 1$ means reward reduction.

3. Quadrant III: $\alpha > 0$ and $\frac{\pi(a_{i,t}|q,a_{i,<t},\theta_{old})}{\mu(a_{i,t}|q,a_{i,<t},\theta_{old})} > 1$. Thus, $\rho_{i,t} > 1$ means punishment enhancement.

4. Quadrant IV: $\alpha > 0$ and $\frac{\pi(a_{i,t}|q,a_{i,<t},\theta_{old})}{\mu(a_{i,t}|q,a_{i,<t},\theta_{old})} < 1$. Thus, $\rho_{i,t} < 1$ means punishment reduction.

Conversely, if $Y < X$, which indicates that the current sequence-level discrepancy is smaller than the reference value, we increase the training-inference discrepancy. The four-quadrants update rule is delineated below (corresponding to Figure 5).

1. Quadrant I: $\alpha > 0$ and $\frac{\pi(a_{i,t}|q,a_{i,<t},\theta_{old})}{\mu(a_{i,t}|q,a_{i,<t},\theta_{old})} < 1$. Thus, $\rho_{i,t} < 1$ means reward reduction.

2. Quadrant II: $\alpha > 0$ and $\frac{\pi(a_{i,t}|q,a_{i,<t},\theta_{old})}{\mu(a_{i,t}|q,a_{i,<t},\theta_{old})} > 1$. Thus, $\rho_{i,t} > 1$ means reward enhancement.

3. Quadrant III: $\alpha < 0$ and $\frac{\pi(a_{i,t}|q,a_{i,<t},\theta_{old})}{\mu(a_{i,t}|q,a_{i,<t},\theta_{old})} > 1$. Thus, $\rho_{i,t} < 1$ means punishment reduction.

4. Quadrant IV: $\alpha < 0$ and $\frac{\pi(a_{i,t}|q,a_{i,<t},\theta_{old})}{\mu(a_{i,t}|q,a_{i,<t},\theta_{old})} < 1$. Thus, $\rho_{i,t} > 1$ means punishment enhancement.

*Remark* 3. In modern LLM post-training scenarios, avoiding an excessive large training-inference discrepancy is more important than avoiding an excessively small training-inference discrepancy, because a large train-inference gap can cause training collapse. Thus, reducing the training-inference discrepancy when $Y > X$ is the primary mechanism because it directly prevents the training from collapsing. Actively increasing the training-inference discrepancy when $Y < X$ serves as a secondary robustness mechanism that prevents the training-inference discrepancy from being excessively small.

### 4.3 Exploration Analysis

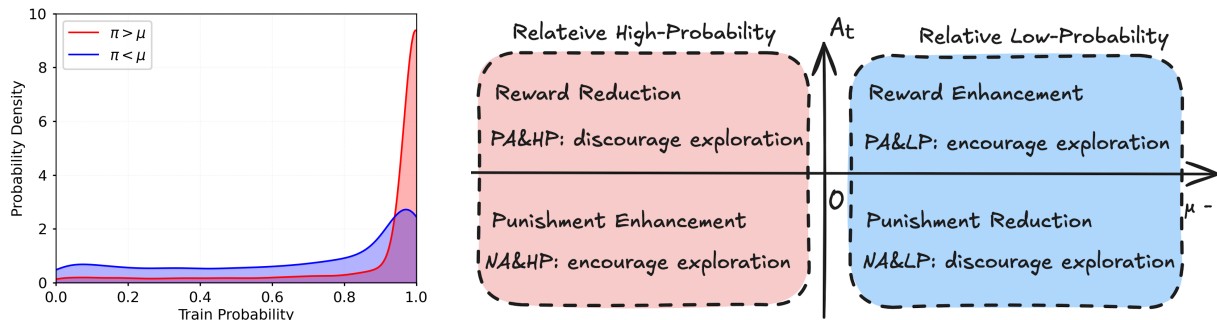

Figure 6: The Probability Density of Training Policy ($\pi$)

Figure 7: AGPA for Steering Exploration

Previous work Su et al. (2025b); Ahmed et al. (2019) links token-level advantage and probability to exploration dynamics: tokens with positive advantage and low probability (PA&LP) and those with negative advantage and high probability (NA&HP) promote exploration(increase entropy), while tokens with negative advantage and low probability (NA&LP) and tokens with positive advantage and high probability(PA&HP) discourage exploration(decrease entropy).

We empirically analyze the distribution of token probabilities using samples from the Qwen2.5-3B Instruct (FP8-quantized) model trained under the VeRL framework, as shown in the probability-density map (Figure 6). While tokens in both the $\pi > \mu$ and $\pi < \mu$ regions can span a wide range of absolute probabilities, the two regions exhibit clear stochastic dominance. Specifically, for any arbitrary absolute probability threshold $P_0 \in (0,1)$, the conditional probabilities satisfy:

$$P(\pi > P_0 \mid \pi > \mu) > P(\pi > P_0 \mid \pi < \mu), \forall P_0 \in (0,1) \tag{13}$$

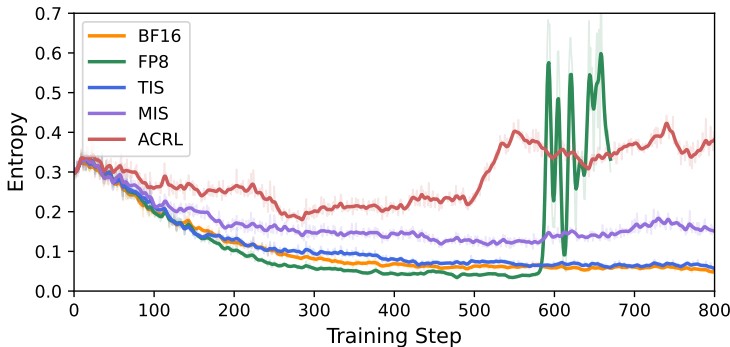

Figure 8: Training entropy for Qwen2.5-3B on GSM8K

and

$$P(\pi \leq P_0 \mid \pi > \mu) < P(\pi \leq P_0 \mid \pi < \mu), \forall P_0 \in (0, 1) \tag{14}$$

This mathematically confirms that the distribution of tokens in the $\pi > \mu$ region is statistically heavily skewed toward higher absolute probabilities compared to the $\pi < \mu$ region. Because our framework operates on the mathematical direction of the probability shift rather than strict absolute boundaries, we reframe these relative probability dynamics. To facilitate further analysis, we explicitly denote the $\pi > \mu$ region as the Relative High-Probability region and the $\pi < \mu$ region as the Relative Low-Probability region.

The AGPA for decreasing the discrepancy(in Figure 3) boosts exploration. As shown in Figure 7, quadrant I and III amplify the contribution of PA&LP and NA&HP tokens in RL training; quadrant II and IV suppress the contribution of PA&HP and NA&LP tokens. Thus, the AGPA for decreasing the discrepancy enhances the impact of tokens that encourage exploration and reduces the impact of tokens that discourage exploration.

In contrast to the increasing discrepancy in direct training Liu et al. (2025a), the training-inference discrepancy under ACRL is constrained within a reasonable range, thereby achieving a net reduction. Combining the net reduction with Figure 7, ACRL promotes exploration and discourages exploitation, thereby improving accuracy.

## 5 Experiments

We design our experiments to comprehensively evaluate the efficacy of ACRL in stabilizing reinforcement learning and improving accuracy under challenging low-precision inference conditions. To systematically validate our claims, we structure our evaluation into six parts: first, we establish ACRL's primary ability to stabilize training and recover accuracy under aggressive quantization (5.1); second, we demonstrate its scalability across larger architectures and different RL algorithms (5.2); third, we analyze its robustness under variations in the reference baseline and extreme discrepancy conditions (5.3); fourth, we examine its sensitivity to the control strength and its computational efficiency (5.4); fifth, we evaluate the benefit of bidirectional discrepancy adjustment through targeted ablation studies (5.5); and finally, we discuss the robustness and scope of our evaluation methodology (5.6).

### 5.1 Stabilizing RL under Aggressive Quantization

To evaluate ACRL's core stabilization capabilities, we conduct experiments across two different scales and difficulty levels: the Qwen2.5-3B-Instruct model evaluated on the GSM8K benchmark Cobbe et al. (2021), and the Qwen2.5-7B-Base model trained on KlearReasoner-MathSub-30K Su et al. (2025a) and evaluated on a suite of difficult mathematical benchmarks (AIME-24, AIME-25, AMC-23, HMMT25, and MATH500).

Table 1: Accuracy of Qwen2.5-3B on GSM8K

| Method | Acc | avg. Acc (last 200 steps) |
|--------|-----|---------------------------|
| BF16 | 87.72% | 86.60% |
| TIS | 88.17% +0.45% | 86.73% +0.13% |
| MIS | 88.17% +0.45% | 86.29% -0.31% |
| ACRL | 88.10% +0.38% | 87.05% +0.45% |

Table 2: Accuracy on GSM8K (Truncated FP8)

| Method | Result | Accuracy |
|--------|--------|----------|
| BF16 | success | 85.90% |
| FP8 | failure | × |
| TIS | success | 84.69% -1.21% |
| MIS | failure | × |
| ACRL | success | 86.13% +0.23% |

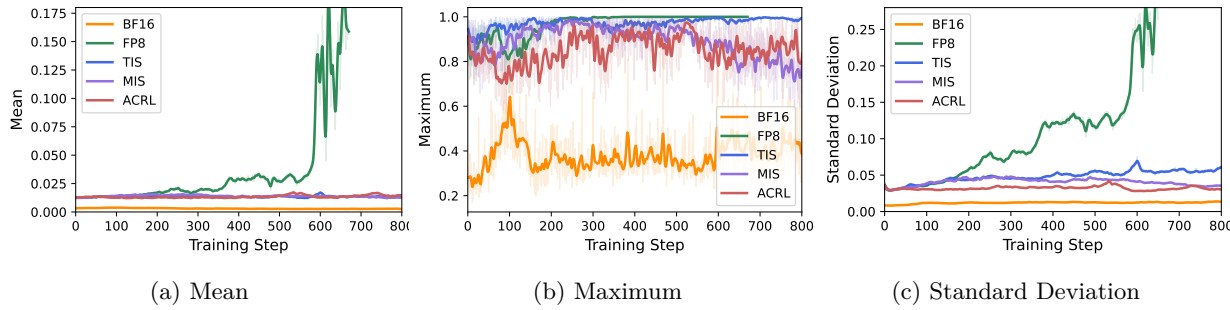

(a) Mean      (b) Maximum      (c) Standard Deviation

Figure 9: The Training-Inference Discrepancy for Qwen2.5-3B on GSM8K

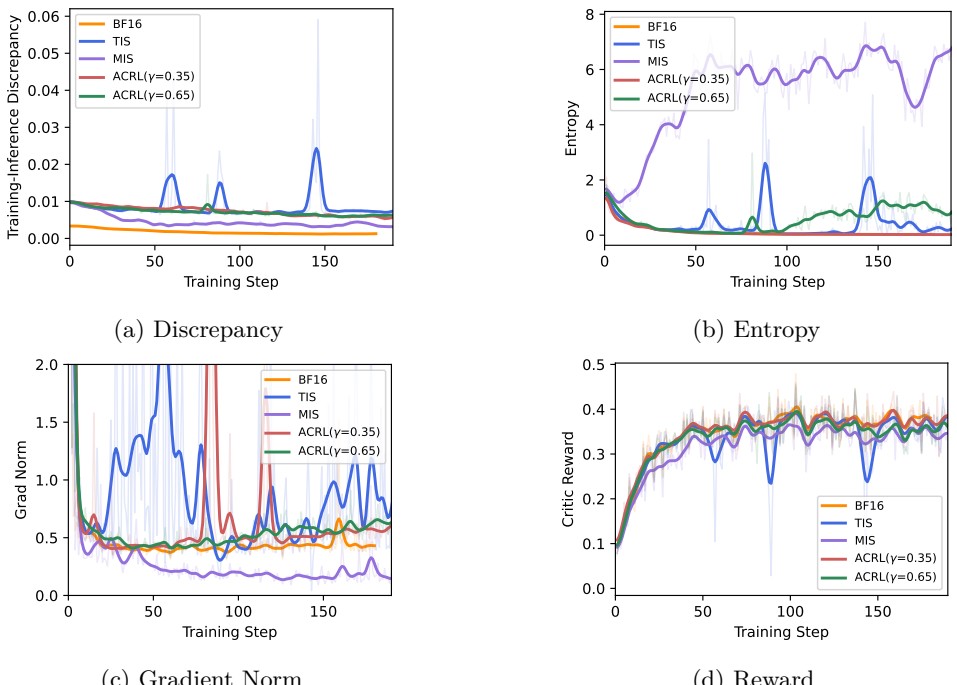

(a) Discrepancy      (b) Entropy

(c) Gradient Norm      (d) Reward

Figure 10: Training-inference discrepancy, entropy, gradient norm, and reward of Qwen2.5-7B on difficult mathematical datasets.

Our experiments are implemented with the VeRL framework Sheng et al. (2025), utilizing vLLM as the inference engine and FSDP as the training engine. We establish a high-precision baseline by maintaining BF16 precision across both engines. To induce training-inference discrepancy, we apply standard nearest-around FP8 quantization (per-token on activations, per-channel on weights) to the inference engine. We compare our proposed ACRL algorithm against the uncorrected FP8 baseline, Token-level Importance Sampling (TIS) Liu et al. (2025b), and Sequence-level Masked Importance Sampling (MIS) Liu et al. (2025a). For ACRL, we

Table 3: Accuracy of Qwen2.5-7B on Difficult Mathematical Datasets

| Method | AIME24 | AIME25 | AMC23 | HMMT25 | MATH500 | Average |
|---|---|---|---|---|---|---|
| BF16 | 16.67% | 13.33% | 65.00% | 3.33% | 77.20% | 35.11% |
| TIS | 13.33% | 16.67% | 62.50% | 3.33% | 77.00% | 34.57% |
| MIS | 20.00% | 13.33% | 60.00% | 3.33% | 78.00% | 34.93% |
| ACRL($\gamma = 0.35$) | 16.67% | 13.33% | 62.50% | 3.33% | 77.40% | 34.65% |
| ACRL($\gamma = 0.65$) | 23.33% | 10.00% | 67.50% | 3.33% | 77.80% | 36.39% |

derive the initial reference discrepancy $X$ from step 0 (yielding $X = 0.013$ for the 3B model and $X = 0.01$ for the 7B model) and apply control strengths of $\gamma = 0.35$ or $\gamma = 0.65$. All correction methods use a clip value of 3.

**Discrepancy Control and Training Stability.** We evaluate the training-inference discrepancy using the distance metric defined in equation 8. As illustrated in Figure 9 (3B model) and Figure 10a (7B model), the uncorrected FP8 baseline suffers from an uncontrolled discrepancy explosion, leading to training collapse. While TIS regulates the discrepancy to some extent, it exhibits high variance and instability, indicated by sharp curve spikes across metrics. ACRL successfully and consistently regulates the discrepancy through the entirety of training, achieving a lower maximum discrepancy and significantly less variance than TIS.

**Exploration Enhancement.** As explained in Section 4.3, controlling the discrepancy dynamically alters policy entropy, thereby influencing exploration. Figure 8 and Figure 10b demonstrate that ACRL maintains the highest functional entropy across configurations. It is worth noting that while MIS occasionally produces high entropy, manual inspection of MIS-generated outputs revealed a preponderance of non-meaningful, low-probability responses. Because MIS generates these "garbage" tokens, its sequence-level importance ratios vanish, leading to vanishing gradients, slow convergence, and low overall reward (Figure 10c,d).

**Accuracy Recovery and Eliminating Low-Precision Quantization Accuracy Degradation.** The ultimate metric of stabilization is the recovery of model accuracy. As summarized in Table 1, ACRL achieves an average pass@1 accuracy of 87.05% over the final 200 steps on GSM8K, outperforming the BF16 baseline and demonstrating superior reliability compared to the erratic peak performance of TIS.

This trend is further magnified on the difficult mathematical datasets with the 7B model (Table 3). While absolute numerical gains on ultra-hard benchmarks like AIME may appear modest (e.g., solving 1-2 additional problems), they represent substantial relative improvements given the extreme difficulty of the 30-question sets. More importantly, by enabling the 7B model to not only survive aggressive FP8 quantization but actually surpass the BF16 baseline's average accuracy (36.39% vs 35.11% with $\gamma = 0.65$), ACRL effectively eliminates the accuracy degradation caused by the use of low-precision inference. This allows practitioners to leverage the massive memory savings and computational speedup of FP8 rollouts without sacrificing reasoning capability.

## 5.2 Scalability and Generalization

To ensure that the stabilization and exploration benefits of ACRL are not artifacts of a specific algorithm, model scale, or task domain, we conducted a comprehensive suite of generalization experiments.

**Algorithmic Generalization (PPO and DAPO).** While GRPO is highly efficient, Proximal Policy Optimization (PPO) remains the foundational standard for LLM alignment, and variations like DAPO are currently among the most widely used methods in practice. To test its ability to generalize across different RL algorithms, we evaluated ACRL ($\gamma = 0.65$) directly on both PPO and DAPO. As shown in Table 4, ACRL successfully maintained discrepancy control and elevated entropy across both methods. On PPO, ACRL exactly matched the high-precision BF16 baseline (87.34%) and surpassed TIS (86.81%). More notably, on

DAPO, the uncorrected FP8 training catastrophically failed, whereas ACRL stabilized the training and achieved the highest overall accuracy (88.63%), outperforming both BF16 and TIS.

Table 4: Accuracy of ACRL across different RL Algorithms (PPO and DAPO)

| Method | PPO Accuracy | DAPO Accuracy |
|---|---|---|
| BF16 | 87.34% | 87.95% |
| FP8 (Uncorrected) | - | Failed |
| FP8 + TIS | 86.81% | 87.26% |
| FP8 + ACRL | **87.34%** | **88.63%** |

**Architectural Scalability (32B and MoE).** As model parameters scale and architectures become more complex, the numerical instability introduced by quantization typically compounds. This is especially true for Mixture-of-Experts (MoE) architectures, where precision-sensitive top-$k$ routing is highly vulnerable to training-inference mismatch. We scaled our evaluation to the Qwen3-32B dense model ($X = 0.016$) and the Qwen3-30B-A3B MoE model ($X = 0.014$), applying ACRL with $\gamma = 0.65$ under GRPO. Detailed compute budgets and training horizons for these experiments are provided in Appendix B. Table 5 demonstrates that ACRL seamlessly scales to 30B+ parameter regimes. In both the dense and MoE architectures, the FP8 ACRL configuration achieved a higher average accuracy over the final 200 steps (steps 600-800) than the high-precision BF16 baseline.

Table 5: Architectural Scalability: Qwen3-32B and Qwen3-30B-A3B (MoE)

| Method | Qwen3-32B (Dense) | | Qwen3-30B-A3B (MoE) | |
|---|---|---|---|---|
| | Peak Acc | Avg Acc (600-800) | Peak Acc | Avg Acc (600-800) |
| BF16 | **0.9674** | 0.9606 | 0.9613 | 0.9558 |
| FP8 + ACRL | 0.9659 | **0.9613** | **0.9621** | **0.9578** |

**Task Transferability (MMLU-Pro).** Finally, to verify that the accuracy gains from ACRL's controlled exploration do not overfit to mathematical reasoning, we extended our evaluation to the MMLU-Pro benchmark. MMLU-Pro consists of 12,000 rigorously curated questions spanning 14 diverse domains, ranging from biology and physics to law and philosophy. Evaluating the Qwen2.5-3B-Instruct models trained in Section 5.1, we found that the underlying reasoning capabilities generalized broadly. As detailed in Table 6, ACRL achieved the highest overall accuracy (0.4534 with FP8 inference; 0.4562 with BF16 inference), outperforming both TIS and the standard BF16 training baseline. For detailed domain scores, please see the appendix. This confirms that the adaptive discrepancy control mechanisms within ACRL yield robust improvements to general model reasoning across various domains beyond math.

Table 6: MMLU-Pro Benchmark Overall Accuracy (14 Domains)

| Training Method | FP8 Inference | BF16 Inference |
|---|---|---|
| BF16 Baseline | 0.4484 | 0.4491 |
| FP8 + TIS | 0.4480 | 0.4530 |
| FP8 + ACRL | **0.4534** | **0.4562** |

### 5.3 Algorithm Robustness and Stress Testing

We systematically stress-test ACRL on the Qwen2.5-3B-Instruct model, showing that it effectively tolerates variations in the initial baseline $X$ and successfully prevents training collapse under extreme truncated FP8 quantization.

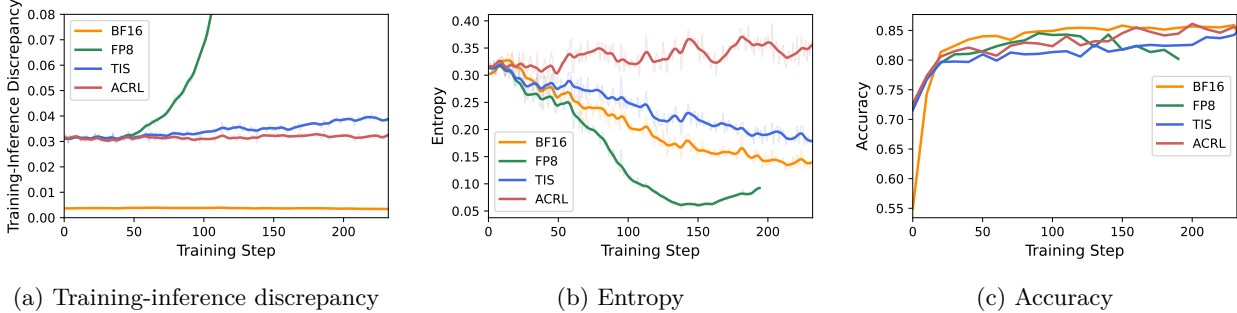

(a) Training-inference discrepancy      (b) Entropy      (c) Accuracy

Figure 11: Training-inference discrepancy, entropy, and accuracy of Qwen2.5-3B on GSM8K (Truncated FP8)

**Robustness of the Reference Baseline ($X$).** ACRL utilizes a static reference baseline $X$ computed at step 0. To evaluate whether ACRL is fragile to the exact initialization of $X$, we tested variations around the accurate baseline ($X = 0.013$). As shown in Table 7, the training remains perfectly stable regardless of minor estimation errors. When $X$ is set lower than the accurate baseline, ACRL further decreases the discrepancy, leading to higher entropy and higher accuracy.

Table 7: Ablation on Reference Baseline Selection ($X$)

| $X$ | Accuracy | Entropy Level |
|---|---|---|
| 0.010 | **0.8757** | High |
| 0.013 | 0.8749 | Medium |
| 0.016 | 0.8680 | Low |

A common concern with a static $X$ is whether the "natural" discrepancy floor shifts dramatically as model weights evolve over thousands of gradient updates. We tracked the step-discrepancy evolution of a standard BF16 GRPO training run measured under FP8 inference. As demonstrated in Table 8, the baseline discrepancy remains remarkably stable (hovering between 0.011 and 0.017) through 900+ steps. This empirical stability validates our use of a static $X$ computed from initial weights.

Table 8: Evolution of Baseline Discrepancy ($X$) Over Training Steps

| Step | 0 | 100 | 200 | 300 | 400 | 500 | 600 | 700 | 800 | 900 | 928 |
|---|---|---|---|---|---|---|---|---|---|---|---|
| Discrepancy | 0.013 | 0.014 | 0.014 | 0.014 | 0.015 | 0.014 | 0.017 | 0.011 | 0.014 | 0.015 | 0.015 |

**Robustness Under Extreme Discrepancy (Stress Testing).** While nearest-around quantization represents standard practice, we further evaluated ACRL's limits by deliberately injecting extreme training-inference discrepancy. We replaced the standard quantization scheme with directed truncation, implemented as $Q(x) = \text{trunc}(x/S)$. This aggressively forces positive values to the next lowest FP8 number and negative values to the next highest, intentionally introducing a much harsher precision mismatch.

For this stress test, TIS, MIS, and ACRL utilized a clip value of 5. For ACRL, we set $\gamma = 0.35$ and a newly computed $X = 0.031$. As shown in Table 2, under these extreme conditions, uncorrected FP8 immediately collapsed, and MIS failed entirely due to vanishing importance sampling ratios (on the order of $10^{-15}$). ACRL, however, successfully clamped the exploding discrepancy, maintained the highest exploration entropy, and

successfully recovered accuracy (86.13%), outperforming both TIS and the BF16 baseline. This demonstrates that ACRL remains robust even when the environment actively induces severe training-inference discrepancy.

## 5.4 Algorithm Sensitivity and Computational Efficiency

To evaluate the practical deployment of ACRL, we investigate two critical operational factors: the algorithm's sensitivity to its primary control hyperparameter and the computational latency it introduces into the training loop.

**Sensitivity to Control Strength ($\gamma$).** The parameter $\gamma$ dictates the magnitude of the discrepancy correction. We ablated $\gamma$ across a spectrum from 0.1 to 2.5 (Table 9). We observed that medium-level control ($\gamma \in [0.5, 1.7]$) yields stable training and optimal accuracy. A severely restricted discrepancy (large $\gamma \geq 2.1$) over-constrains the optimization space; it forces the training policy to overfit the inference policy, limiting viable weight selection and eventually causing training collapse in later steps. Conversely, weak control ($\gamma = 0.1$) allows the discrepancy to grow noticeably larger than optimal, risking instability over longer training horizons.

Table 9: Sensitivity Analysis of Control Strength ($\gamma$) on GSM8K

| Control Strength ($\gamma$) | Accuracy | Completion |
|---|---|---|
| 0.1 | 0.8749 | Success |
| 0.5 | 0.8726 | Success |
| 0.9 | 0.8787 | Success |
| 1.3 | **0.8832** | Success |
| 1.7 | 0.8802 | Success |
| 2.1 | 0.8696 | Failure |
| 2.5 | 0.8741 | Failure |

**Computational Overhead.** A practical concern when introducing adaptive control mechanisms is the potential for training latency. However, ACRL incurs negligible computational overhead. Similar to standard importance sampling, the information required for ACRL (the rollout probabilities) is naturally computed during the inference stage and cached for later use during the training stage.

To precisely quantify this, we profiled the training wall-time of the Qwen2.5-3B-Instruct model using GRPO with ACRL and FP8 quantization. In our environment, a single RL step requires an average of 56.496 seconds. The isolated ACRL function call—which computes the importance sampling weight tensor from the cached log probabilities—takes only 0.059 seconds. This introduces an execution overhead of just $\frac{t_{\text{ACRL}}}{t_{\text{step}}} \times 100\% = \frac{0.059\text{s}}{56.496\text{s}} \times 100\% \approx 0.1\%$. Considering that rollout inference dominates the time within a single RL step, the additional computational cost of achieving stability through ACRL is practically unnoticeable.

## 5.5 Ablation of Bidirectional Discrepancy Control

To evaluate the benefit of actively increasing the discrepancy when $Y < X$, we compare full ACRL against two one-sided variants using the Qwen2.5-3B-Instruct model on GSM8K, as summarized in Table 10.

**Continuous Reduction.** This variant removes the $Y < X$ reference-restoring branch and continuously pushes the discrepancy toward zero, even when $Y < X$. Specifically, we fix $\alpha = -1$ in Quadrants I and II and $\alpha = 1$ in Quadrants III and IV, so that the update always reduces the discrepancy rather than reversing direction when it falls below the reference value $X$. With a more aggressive setting of $\alpha = -3$ in Quadrants I and II and $\alpha = 3$ in Quadrants III and IV, training immediately collapses. With the standard setting of $\alpha = \pm 1$, training completes but underperforms full ACRL by 0.91 percentage points in both peak accuracy and average accuracy over the final 200 steps.

**Fallback to GRPO.** This variant disables ACRL when $Y < X$ and instead reverts to standard GRPO updates. Although this variant completes training, it exhibits a pronounced crash-and-recovery pattern

between steps 420 and 460 and underperforms full ACRL by 0.38% in peak accuracy and 0.08% in average accuracy over the final 200 steps. These results demonstrate that neither continuously reducing the discrepancy nor passively ignoring the $Y < X$ regime achieves the stability and performance of full ACRL, supporting the benefit of bidirectional discrepancy control for maintaining the discrepancy near the reference value.

Table 10: Ablation Results for One-Sided Discrepancy-Control Variants

| Method | Accuracy | Avg. Accuracy (Last 200 Steps) |
|---|---|---|
| BF16 Baseline | 87.72% | 86.60% |
| Full ACRL | **88.10%** | **87.05%** |
| ACRL Continuous Reduction | 87.19% | 86.14% |
| ACRL Fallback to GRPO | 87.72% | 86.97% |

### 5.6 Evaluation Robustness and Scope

**Evaluation Robustness.** Conducting multiple independent training runs for every model and baseline is computationally prohibitive at the 3B, 7B, and 32B scales. To reduce sensitivity to single-step fluctuations, we report the average pass@1 accuracy over the final 200 training steps after convergence rather than relying solely on peak accuracy. We further evaluate ACRL across multiple model scales (3B, 7B, and 32B), architectures (Dense and MoE), RL algorithms (GRPO, PPO, and DAPO), and benchmark suites (Math and MMLU-Pro). Although temporal averaging does not replace independent-seed variance, the consistent recovery of accuracy to levels matching or exceeding the BF16 baselines, together with the stable training observed across these diverse settings, provides broad empirical evidence for the robustness of ACRL.

**Evaluation Scope.** The empirical evaluation in this study focuses on training-inference discrepancies introduced by low-precision FP8 inference in single-turn reasoning tasks. Specifically, we evaluate ACRL using vLLM as the inference engine and FSDP as the training engine. Although the ACRL formulation is designed to control general training-inference discrepancies, the current experiments do not comprehensively evaluate high-precision backend mismatches, multi-turn or tool-integrated RL, asynchronous rollout and training pipelines, or alternative combinations of training and inference engines. We therefore restrict our empirical claims to the evaluated FP8 single-turn setting and leave these broader scenarios for future work.

## 6 Limitations

While ACRL effectively stabilizes RL training under high training-inference discrepancy, our study has several limitations that warrant future investigation. First, regarding hyperparameter sensitivity, our method relies on the control strength $\gamma$ and a static reference discrepancy $X$. While our empirical results show stability across a reasonable range of $\gamma$, extreme values can constrain the optimization space or provide insufficient control. Furthermore, $X$ is computed from the initial training step; although we observed that the natural discrepancy floor remains relatively stable over our training horizons, it is possible that for exceptionally long training runs where model weights evolve drastically, a dynamic or moving-average $X$ might be required to maintain optimal stability.

Second, our empirical validation primarily focuses on mathematical reasoning tasks and the vLLM/FSDP backend combination. While we demonstrated task transferability on MMLU-Pro and algorithm transferability across PPO and DAPO, the exploration benefits of ACRL have not yet been comprehensively validated in other RLHF domains with highly subjective rewards, such as creative writing, coding, or dialogue safety. Finally, while architectural separation and quantization are universal sources of discrepancy, future work should systematically evaluate ACRL across a wider variety of emerging training and inference backend pairings.

# 7 Conclusion

In this paper, we address the training-inference discrepancy, a critical issue that can cause instability and lead to training collapse in RL. We propose the ACRL algorithm from a novel perspective, enabling stable reinforcement learning by adaptively controlling this discrepancy. By maintaining the mismatch within a reasonable range, ACRL not only stabilizes the optimization process but also increases accuracy by inherently enhancing exploration. Our experiments demonstrate that our approach outperforms TIS in stability, achieves a convergence rate superior to MIS, and matches or exceeds the accuracy of high-precision BF16 baselines across multiple RL algorithms. Future work may explore the impact of alternative discrepancy measurements, such as KL divergence, investigate dynamic baseline scaling for exceptionally long training horizons, and evaluate ACRL's exploration benefits within highly subjective RLHF domains.

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

# A   Hyper-parameters used for experiments training.

Table 11: Hyper-parameters used for Qwen2.5-3B Instruct on GSM8K

| | |
|---|---|
| data.train__batch__size | 64 |
| data.max__response_length | 2048 |
| actor__rollout__ref.actor.optim.lr | 5e-7 |
| actor__rollout__ref.actor.entropy__coeff | 0.0 |
| actor__rollout__ref.actor.ppo__mini__batch__size | 64 |
| actor__rollout__ref.actor.ppo__micro__batch__size__per__gpu | 1 |
| actor__rollout__ref.actor.kl__loss__coef | 0 |
| actor__rollout__ref.actor.grad__clip | 1 |
| actor__rollout__ref.rollout.log__prob__micro__batch__size__per__gpu | 1 |
| actor__rollout__ref.rollout.name | vllm |
| actor__rollout__ref.rollout.gpu__memory__utilization | 0.6 |
| actor__rollout__ref.rollout.n | 8 |
| actor__rollout__ref.ref.log__prob__micro__batch__size__per__gpu | 1 |
| algorithm.kl__ctrl.kl__coef | 0 |
| trainer.critic__warmup | 0 |
| trainer.test__freq | 10 |

Table 12: Hyper-parameters used for Qwen2.5-7B Base on Difficult Mathematical Datasets

| | |
|---|---|
| data.train__batch__size | 128 |
| data.max__response_length | 8192 |
| actor__rollout__ref.actor.optim.lr | 1e-6 |
| actor__rollout__ref.actor.entropy__coeff | 0.001 |
| actor__rollout__ref.actor.ppo__mini__batch__size | 16 |
| actor__rollout__ref.actor.ppo__micro__batch__size__per__gpu | 1 |
| actor__rollout__ref.actor.kl__loss__coef | 0 |
| actor__rollout__ref.actor.grad__clip | 1 |
| actor__rollout__ref.rollout.log__prob__micro__batch__size__per__gpu | 1 |
| actor__rollout__ref.rollout.name | vllm |
| actor__rollout__ref.rollout.gpu__memory__utilization | 0.6 |
| actor__rollout__ref.rollout.n | 8 |
| actor__rollout__ref.ref.log__prob__micro__batch__size__per__gpu | 1 |
| actor__rollout__ref.rollout.max__num__batched_tokens | 32768 |
| algorithm.kl__ctrl.kl__coef | 0 |
| trainer.critic__warmup | 0 |
| trainer.test__freq | 10 |

## B    Compute Budgets for the 32B Dense and 30B MoE Models

We disclose the compute budgets used for the Qwen3-32B (Dense) and Qwen3-30B-A3B (MoE) experiments. Both models were trained using four NVIDIA B200 GPUs with 180 GB of memory per GPU for a total of 800 steps. The Qwen3-32B Dense model required 21 hours, while the Qwen3-30B-A3B MoE model required 46.5 hours, or approximately 1.94 days. The longer training time of the MoE model resulted from the conservative inference-concurrency configuration used for that run.

Both models reached convergence at approximately step 400 and were trained until step 800 to evaluate post-convergence stability. At step 400, ACRL achieved an accuracy of 95.51% on the Qwen3-32B Dense model, closely matching the 95.53% BF16 baseline. On the Qwen3-30B-A3B MoE model, ACRL achieved 96.21%, compared with 95.22% for the BF16 baseline. The remaining training steps showed no observed collapse or sustained oscillatory degradation.

## C MMLU-Pro: detailed results across domains.

Table 13: Detailed MMLU-Pro Evaluation Scores across 14 Domains (FP8 Inference)

| Task Category | BF16 Baseline | FP8 + TIS | FP8 + ACRL |
|---|---|---|---|
| Biology | 0.5997 | 0.5955 | **0.6332** |
| Business | 0.5336 | **0.5425** | 0.5361 |
| Chemistry | **0.4320** | 0.4019 | 0.4187 |
| Computer Science | 0.4683 | 0.4659 | **0.4756** |
| Economics | 0.5450 | 0.5427 | **0.5498** |
| Engineering | 0.3034 | 0.3168 | **0.3251** |
| Health | 0.4218 | **0.4279** | 0.4267 |
| History | 0.3780 | 0.3727 | **0.3858** |
| Law | 0.2425 | **0.2489** | 0.2425 |
| Math | 0.5759 | **0.5811** | 0.5633 |
| Philosophy | **0.3828** | 0.3747 | 0.3647 |
| Physics | 0.4296 | 0.4349 | **0.4565** |
| Psychology | 0.5602 | 0.5589 | **0.5727** |
| Other | **0.4102** | 0.4058 | 0.4048 |
| **Overall** | 0.4484 | 0.4480 | **0.4534** |

Table 14: Detailed MMLU-Pro Evaluation Scores across 14 Domains (BF16 Inference)

| Task Category | BF16 Baseline | FP8 + TIS | FP8 + ACRL |
|---|---|---|---|
| Biology | 0.5802 | 0.5983 | **0.6081** |
| Business | 0.5526 | **0.5602** | 0.5399 |
| Chemistry | 0.4364 | 0.4337 | **0.4461** |
| Computer Science | **0.4439** | 0.4366 | 0.4220 |
| Economics | 0.5273 | 0.5438 | **0.5604** |
| Engineering | 0.3148 | 0.3096 | **0.3488** |
| Health | 0.4254 | **0.4364** | 0.4218 |
| History | 0.3858 | 0.3885 | **0.4094** |
| Law | **0.2498** | 0.2389 | 0.2216 |
| Math | 0.5655 | 0.5751 | **0.5781** |
| Philosophy | 0.3768 | 0.3868 | **0.3928** |
| Physics | 0.4426 | **0.4534** | 0.4457 |
| Psychology | 0.5652 | 0.5576 | **0.5764** |
| Other | 0.4080 | **0.4091** | 0.4080 |
| **Overall** | 0.4491 | 0.4530 | **0.4562** |

