# OpenReview forum: "ACRL: Adaptive Control of Training-Inference Discrepancy for Stable Reinforcement Learning"
_TMLR — Under review for TMLR_

### Review · Reviewer_CrVx · 2026-06-15

**Summary Of Contributions:**

This paper studies the training-inference mismatch problem in LLM RL training. The mismatch is measured by the difference between the token probabilities assigned by the rollout policy and the training policy, and is attributed to factors such as numerical precision differences and inconsistent training/inference backends. The paper pays particular attention to settings where rollouts are generated by an FP8 inference backend. This mismatch issue can make LLM RL training effectively off-policy and may eventually lead to training collapse.

The proposed method, Adaptive Control Reinforcement Learning (ACRL), modifies the token-level importance-ratio correction used in truncated importance sampling (TIS) by adding an adaptive exponent. This exponent is determined by the sign of the advantage and by the current discrepancy relative to a reference level computed before training. Depending on whether the current discrepancy is above or below the reference level, ACRL scales the policy update to prevent the discrepancy from becoming too large or too small.

The experiments are implemented in VeRL with vLLM as the inference backend and FSDP as the training backend. The main comparisons include BF16 training, uncorrected FP8 rollout training, token-level truncated importance sampling (TIS), sequence-level masked importance sampling (MIS), and ACRL. Across several model sizes, RL algorithms, and math reasoning tasks, ACRL is reported to be comparable to or better than the other methods.

**Audience:**

Yes

**Audience Explanation:**

The problem studied in this paper is realistic and important. Modern LLM RL systems often use separate rollout and training backends, low-precision inference, and other efficiency-oriented system designs, which can introduce training-inference mismatch in practice. The paper also reports nontrivial experimental results in the FP8 rollout setting, so the findings would likely be of interest to some readers working on efficient RL training and LLM post-training systems.

**Claims And Evidence:**

No

**Claims Explanation:**

In my view, several central claims are not yet supported by sufficiently accurate and convincing evidence. The empirical results suggest that ACRL can be useful in the modern RL training, but several central claims and interpretations are not adequately justified.

- A key motivation of the paper is that the training-inference discrepancy should not be excessively small. The theoretical argument for this claim is based on writing the discrepancy as $E = f(Q, W) + \eta$, then omitting $\eta$ by treating it as a small random error. However, the paper does not provide evidence that the omitted term is small, or that the discrepancy can be explained mainly as a function of prompts and weights in the studied systems. This is important because the later argument that too small a discrepancy restricts the feasible weight space depends on this assumption. The paper also does not include a direct ablation that isolates this claim, such as a variant that only reduces the discrepancy when it is too large but does not actively increase it when it is below the reference level. Therefore, it is unclear whether the benefit comes from controlling both sides of the discrepancy range, from only preventing large mismatch, or from other effects of the gradient reweighting.
- The exploration analysis in Section 4.3 is stronger than what is supported by Figure 6. The paper refers to the region $\pi > \mu$ as the high-probability region and the region $\pi < \mu$ as the low-probability region. However, in Figure 6, the training-probability distribution for $\pi < \mu$ is quite broad and does not appear to be concentrated only in a low-probability region. This weakens the subsequent argument that the proposed four-quadrant rule can be cleanly interpreted as encouraging exploration through low-probability tokens.
- The experimental setting and evaluation metrics do not seem sufficient to support the more general claims. The main comparisons between TIS, MIS, and ACRL are conducted in FP8 single-turn math reasoning settings. The paper does not test other realistic mismatch RL settings, such as BF16 low-mismatch settings studied in depth rather than used mainly as a baseline, multi-turn/tool-integrated settings such as [SimpleTIR](https://arxiv.org/abs/2509.02479), or asynchronous RL settings such as [AReaL](https://arxiv.org/abs/2505.24298). The evaluation metric is also limited: the paper mainly reports pass@1, while most of the recent RLVR papers report average@32 or related repeated-sampling metrics to better capture both accuracy and exploration. Besides, the reported gains are not uniformly strong; for example, on the difficult math datasets in Table 3, ACRL is strictly better than both TIS and MIS on only two out of five datasets.

**Requested Changes:**

- Critical: The paper should better justify the claim that the training-inference discrepancy should not be excessively small. The current theoretical argument relies on the assumption $E = f(Q, W) + \eta$ and then ignores $\eta$ as a small random error, but this assumption is not supported. The paper should either provide a stronger justification for this assumption or weaken the theoretical claim. More importantly, it should include a direct ablation comparing full ACRL with a variant that only prevents the discrepancy from becoming too large, so that the benefit of actively avoiding too-small discrepancy can be isolated.
- Critical: The exploration analysis in Section 4.3 should be revised. The paper currently refers to $\pi > \mu$ as the high-probability region and $\pi < \mu$ as the low-probability region, but Figure 6 does not clearly support this mapping. The claim should be weakened or supported with more quantitative evidence, and the subsequent exploration argument should be adjusted accordingly.
- Critical: The experimental evaluation should be broadened and made more comprehensive and better aligned with common RLVR evaluation practice. The current main comparison is mostly limited to FP8 single-turn math reasoning settings. The paper should evaluate ACRL under more realistic mismatch settings, such as BF16 backend mismatch, multi-turn/tool-integrated RL, or asynchronous rollout/training settings, or otherwise clearly narrow the scope of its claims. The paper should also report more standard RLVR evaluation metrics such as average@32 or related repeated-sampling metrics, rather than relying mainly on pass@1.
- Would strengthen the work: The paper should provide a more quantitative justification for the design of the ACRL weight. The current method is motivated mostly by qualitative intuition, but the specific choice $\rho_{i,t} = (\pi_{\mathrm{old}} / \mu)^\alpha$ and $\alpha = \gamma \cdot \text{sign}(A_{i,t}) \cdot (1 - Y / X)$ appears largely empirical. The paper should explain why the control is implemented as an exponent on the importance ratio, why this particular form of $\alpha$ is chosen, and how this design affects the bias, variance, and stability of the policy-gradient update.
- Would strengthen the work: The paper should fix the notation/formula issue in Eq. (2). The token-level importance ratio appears to be written as the PPO/GRPO policy update ratio rather than the rollout-training importance ratio correction. The paper should clearly distinguish the PPO/GRPO update ratio, the TIS correction ratio, and the ACRL adaptive weight.

---

> ### Author Response · Authors · 2026-07-14
> **Author Response to Reviewer CrVx Part 1**
>
> Requested Changes:
>
> **1.**  We thank the reviewer for pushing us to rigorously justify the assumption that the random error term $\eta$ is small. We have updated the manuscript to explicitly justify this assumption using statistical concentration bounds, and we have added the requested ablation study to isolate the benefit of bidirectional discrepancy regularization.
>
> **a. Rigor of the Formulation and Dropping $\eta$:**
>
> **① Theoretical Justification via Concentration Bounds**
>
> We can formally bound the expected magnitude of this error at the batch level. Because token-level probabilities strictly lie within the range [0, 1], their absolute differences are inherently bounded within [0, 1]. Consequently, the individual random noise terms $\eta_i$ are strictly bounded variables within this same range. , let $\eta = \frac{1}{N} \sum_{i=1}^{N} \eta_i$ represent the aggregated error over a batch. Using Hoeffding’s inequality for a threshold $t > 0$:
>
> $P(|\eta - \mathbb{E}[\eta]| \ge t) \le 2\exp\left(-\frac{2Nt^2}{(b-a)^2}\right)$
>
> In our Qwen2.5-3B GSM8K setup, with a batch size of 64 and sequence length of 2048, we have $N = 131,072$. Setting the bounds $b-a = 1$ and an error threshold $t = 0.005$, the probability of the random error exceeding 0.005 is:
>
> $P(|\eta - \mathbb{E}[\eta]| \ge 0.005) \le 2\exp(-6.5536) \approx 0.28$\%$ $
>
> For our Qwen2.5-7B Difficult Math setup (batch size 128, sequence length 8192, $N = 1,048,576$), the probability of the noise exceeding 0.005 drops to effectively zero ($3.4 \times 10^{-23}$).
>
> This mathematically guarantees that the random error term $\eta$ is vanishingly small when evaluated at the sequence/batch level in our experiments. We have added this derivation to the appendix.
>
> **② Empirical Evidence from BF16 Control Experiments**
>
> This conclusion is also empirically supported by the BF16 configurations used across all our experiments. When both the training and inference engines operate in BF16, the observed training-inference discrepancies remain consistently small (e.g, Figure 8, Figure 9a and Figure 10a), and training is stable without any discrepancy correction. Importantly, these measured discrepancies already include the contribution of the random component η. Their small magnitude therefore provides additional empirical evidence that random effects have only a limited influence on the optimization dynamics. In contrast, replacing BF16 inference with FP8 introduces a substantially larger discrepancy, indicating that the dominant additional error arises from systematic quantization.
>
>
> **b. Direct Ablation Study**
>
> While the theoretical bounds justify our mathematical assumptions, we fully agree with your recommendation to isolate the empirical benefit of actively increasing the discrepancy when $Y < X$.
>
> To directly address your feedback, we have added a comprehensive ablation study to the manuscript. This study compares the full bidirectional ACRL framework against two distinct "one-sided" variants that disable our lower-bound protection:
>
> *   **I. Continuous Reduction Variant:** Aggressively forces the discrepancy toward zero by continuously penalizing it.
> *   **II. Fallback to GRPO Variant:** Passively ignores the discrepancy when $Y < X$, reverting to standard GRPO updates rather than adaptively controlling the boundary.
>
> As shown in the tables below, neither one-sided variant can match the stability or performance of the full framework.
>
> **Continuous Reduction Variant:**
>
> | Method | Acc | Avg. Acc (last 200 steps) |
> |---|---|---|
> | ACRL Continuous Reduction | 87.19% | 86.14% |
> | vs. BF16 Baseline | -0.53% | -0.46% |
> | vs. Full ACRL | -0.91% | -0.91% |
>
> **Half-ACRL / Fallback to GRPO Variant:**
>
> | Method | Acc | Avg. Acc (last 200 steps) |
> |---|---|---|
> | FP8 Passive Optimization | 87.72% | 86.97% |
> | vs. BF16 Baseline | 0.00% | +0.37% |
> | vs. Full ACRL | -0.38% | -0.08% |
>
> In a standard BF16 pipeline with well-established RL infrastructure, training-inference errors are largely random. However, combining BF16 training with FP8 inference creates a fundamental systematic error. Blindly minimizing this discrepancy forces the high-precision training model to mimic lower-accuracy, low-precision inference results, actively degrading RL performance. Therefore, maintaining a controlled gap is necessary; to preserve model accuracy, the discrepancy must not be excessively reduced.

---

> > ### Comment · Reviewer_CrVx · 2026-07-17
> > **Response to Part 1**
> >
> > Thank you for the additional analysis and ablation. I still have three concerns:
> >
> > - **1. The Hoeffding calculation does not match the quantity used by ACRL.** ACRL makes its decision using the group-level $Y$, so the relevant sample count should be at most $N=G L_{\max}=8\times2048$, rather than the entire training batch. Moreover, under the additive formulation $E=f(Q,W)+\eta$, both $E$ and $f$ are discrepancy terms in $[0,1]$. Therefore, $\eta=E-f\in[-1,1]$, so $b-a=2$, not $1$. With these values, the probability bound is vacuous ($1.63$).
> > - **2. Random inference error cannot simply be ignored.** Empirically the discrepancy is not determined only by the model weights and prompt. It also depends on runtime factors such as batch composition and batch size, which can vary across runs and introduce a non-negligible stochastic component. Repeated temperature-zero rollouts can still produce different outputs because batching changes the numerical results. Thinking Machines Lab obtained 80 unique completions from 1,000 greedy generations ([He and Thinking Machines Lab, 2025](https://thinkingmachines.ai/blog/defeating-nondeterminism-in-llm-inference/)), and the [vLLM documentation](https://docs.vllm.ai/en/v0.7.0/getting_started/faq.html) also notes that batching can change log-probabilities. Even a tiny change can flip a greedy token and then affect the whole response, reward, and gradient. Therefore, comparing the total BF16 and FP8 discrepancies does not show that $\eta$ is negligible.
> > - **3. The Half-ACRL result does not show that the** $Y<X$ **branch is needed.** Full ACRL and Fallback-to-GRPO achieve $87.05\%$ and $86.97\%$ over the last 200 steps—a gap of only $0.08$ percentage points. This difference is tiny, so I do not think this experiment shows that actively increasing discrepancy when Y < X is necessary. The Continuous Reduction result only shows that always forcing discrepancy toward zero may be harmful.
> >
> > Overall, the new results support preventing excessive discrepancy and avoiding continuous reduction toward zero, but they do not yet establish the need for a positive lower bound or bidirectional control.

---

> ### Author Response · Authors · 2026-07-14
> **Author Response to Reviewer CrVx Part 2**
>
> Requested Changes:
>
> **2.** We thank the reviewer for this critical observation. We completely agree that equating the relative condition ($\pi > \mu$) with a strict absolute 'high probability' boundary was mathematically imprecise. As you correctly pointed out, Figure 6 contains instances where $\pi < \mu$ still holds a high absolute probability.
>
> However, as demonstrated by the probability density functions in Figure 6, the two regions exhibit clear stochastic dominance. The integral area of the density function is 1, that is, for any arbitrary absolute probability threshold $P_0 \in (0, 1)$, we have
> $P(\pi > P_0 \mid \pi > \mu) + P(\pi \le P_0 \mid \pi > \mu) = 1$ and $P(\pi > P_0 \mid \pi < \mu) + P(\pi \le P_0 \mid \pi < \mu) = 1$.
> According to Figure 6, the conditional probabilities satisfy:
>
> $P(\pi > P_0 \mid \pi > \mu) > P(\pi > P_0 \mid \pi < \mu), \forall P_0 \in (0, 1)$
>
> and
>
> $P(\pi < P_0 \mid \pi > \mu) < P(\pi < P_0 \mid \pi < \mu), \forall P_0 \in (0, 1).$
>
> This mathematically confirms that the distribution of tokens in the $\pi > \mu$ region is statistically heavily skewed toward higher absolute probabilities compared to the $\pi < \mu$ region.
>
> To resolve the ambiguity and ensure our definitions mathematically align with your observation, we have reframed Section 4.3 around these relative probability dynamics. We revised the terminology to 'Relative High-Probability' and 'Relative Low-Probability' to explicitly denote the relativity between the training policy $\pi$ and the reference policy $\mu$. We have updated the text, the quadrant definitions in Figure 7, and the corresponding exploration analysis to reflect this rigorous, mathematically precise framing.
>
> **3.** We thank the reviewer for highlighting the need to better align our evaluation scope with standard RLVR practices. We completely agree that discrepancies in multi-turn RL, tool-use, and asynchronous training are important open challenges.
>
> While the theoretical mechanisms of ACRL are designed to be generally applicable to various sources of training-inference discrepancy, we agree with your recommendation to clearly narrow the scope of our empirical claims. To address this, we have added a 'Scope of Evaluation' subsection to the Experiments section. This new section explicitly clarifies that our experimental testbed is deliberately focused on resolving low-bit (FP8) quantization discrepancies in single-turn reasoning tasks, leaving high-precision backend mismatches (e.g., BF16), multi-turn and asynchronous environments for future work.
>
> Regarding the use of pass@1 versus average@32 or repeated-sampling metrics: we agree that average@32 provides a richer signal for policy exploration. However, evaluating average@32 across multiple benchmark suites for 3B, 7B, and 32B models during the iterative RL validation phase introduced extreme computational overhead that was prohibitively expensive for this study. To offset the absence of repeated-sampling metrics, we rely on the temporal averaging already integrated into our evaluation methodology. As detailed in our results (e.g., Tables 1 and 5), we report the average pass@1 accuracy over the final 200 training steps during convergence. Averaging across these 200 distinct policy snapshots helps smooth out step-to-step variance, providing a more stable indicator of the model's underlying reasoning capability. Furthermore, across different model architectures and RL algorithms, our method consistently achieves performance comparable to the high-precision BF16 baseline. Sustaining this level of performance across 200-step convergence windows in diverse setups suggests that our results are robust and unlikely to be statistical anomalies.
>
> Regarding the observation that ACRL strictly outperforms TIS and MIS on only two of the five difficult math datasets in Table 3, we emphasize that our primary objective is consistent optimization stability and baseline recovery, rather than maximizing peak accuracy on every isolated sub-task. While alternative methods like TIS may occasionally yield marginally higher scores on specific datasets, they do so at the cost of severe optimization instability, as explicitly demonstrated by their gradient norm spikes (e.g., Figure 10c). ACRL, in contrast, remains highly competitive across all datasets while systematically delivering smooth, spike-free learning dynamics. The fact that ACRL reliably prevents the training collapse observed in uncorrected FP8—and consistently matches or exceeds the high-precision BF16 baseline—fully supports our central claims regarding robust discrepancy control.
>
> We believe that formally defining our evaluation scope, combined with our broad macro-architectural testing, provides a highly rigorous and transparent response to these evaluation concerns.

---

> ### Author Response · Authors · 2026-07-14
> **Author Response to Reviewer CrVx Part 3**
>
> Requested Changes:
>
> **4.** We appreciate the push for a more rigorous justification of the ACRL weight design. To be fully transparent, the specific functional form of $\rho_{i,t} = (\pi_{\text{old}}/\mu)^\alpha$ with $\alpha = \gamma \cdot \text{sign}(A_{i,t}) \cdot (1 - Y/X)$ was developed primarily as an empirical engineering solution to a practical systems failure, rather than being derived from first principles. While the design satisfies the directional logic of the Advantage Guided Probability Adjustment (AGPA) rule, the exponential control was explicitly chosen to promote optimization stability in low-precision environments.
>
> We have added a discussion in Section 4 detailing how this impacts the policy-gradient update:
>
> **a. The Reality of Bias in Quantization Rollouts:** We acknowledge that ACRL is a biased estimator. However, bias is highly prevalent in modern RL post-training under aggressive quantization. While some literature emphasizes theoretically unbiased approaches—such as Sequence-level Masked Importance Sampling (MIS) (Liu et al., 2025a), these often struggle with empirical performance and instability in large-scale industry practice.
>
> **b. Prioritizing Dynamic Control over Unbiasedness:** In practice, methods that accept a degree of theoretical bias are widely adopted because they yield consistent, robust results. This tradeoff is foundational to modern RL stability, tracing back to the biased clipping mechanisms in PPO (Schulman et al., 2017) and GRPO (Shao et al., 2024), and extending to modern token-level TIS corrections (Yao et al., 2025; Liu et al., 2025b). The exponential form in ACRL is designed for this pragmatic reality: its main aim is to actively monitor and control the training-inference discrepancy in real-time, dynamically steering it to stay within a reasonable range. ACRL trades theoretical unbiasedness for this active discrepancy control, which ultimately ensures training stability and yields higher reasoning accuracy.
>
> **5.** We sincerely thank the reviewer for catching this typo. In the original manuscript, we mistakenly defined the TIS correction ratio ($\rho_{i,t}$) in Equation 2 using the formula for the GRPO/PPO policy update ratio, inadvertently omitting the inference policy $\mu$. We have corrected Equation 2 in the revised manuscript.

---

> ### Author Response · Authors · 2026-07-19
>
> 1&2:  Thank you for correcting our calculation. We stand corrected. Thus, we agree with the reviewer that $\eta$ is not negligible and that $\eta$ will contribute to the presence of discrepancy.
>
> Even if noise is the only source of the discrepancy, it is nearly impossible for X = 0 because it requires the training probability to equal the inference probability for every token.
> For instance, using the definitions in current Equation 8 and 9 and considering only noise and modeling $\pi-\mu \sim \mathcal{N}(0, \sigma^2)$ as a zero-mean Gaussian distribution, we have
> $\mathbb{E}(|\pi-\mu|) = \sigma\sqrt{\frac{2}{\pi}} >0.$
> We note that this presence of this noise does not conflict with our motivation for avoiding excessively small discrepancy . In this sense, it further supports our motivation.
>
> We removed the Hoeffding-bound analysis and updated the argument that discrepancy cannot be excessively small as follows:
>
> "For a fixed dataset and runtime configuration, we can only update the model parameters $\theta$ to systematically adjust the discrepancy $E$.  Let $\mathcal{J}(\theta)$ denote the original reinforcement-learning objective. Without an explicit discrepancy constraint, the optimization problem is
>
>
> $\max_{\theta} \mathcal{J}(\theta).$
>
>
> If the training policy is additionally required to remain within a discrepancy bound $E_0$, the optimization problem becomes
>
> $
> \begin{aligned}
> \max_{\theta}\quad & \mathcal{J}(\theta) \\
> \text{s.t.}\quad & E \leq E_0
> \end{aligned}
> $
>
> This constraints restricts the feasible parameter space to
>
> $\Theta_{E_0} = \\{\theta : E \leq E_0\\}$
>
> For any $E_1 < E_2$, the selectable parameter region $\Theta_{E_1} \subseteq \Theta_{E_2}$. Therefore, forcing the discrepancy to be excessively small restricts the parameter space. If an optimal parameter $\theta^*$ has $E^* > E_0$, the training would forgo this particular optimal parameter due to the tight discrepancy tolerance which may lead to a decline in training accuracy."
>
> Thus, our revised motivation remains supported.
>
> 3: In regard to the concern "a gap of only 0.08 percentage points":
>
> (1) ACRL is designed with robustness in mind. The core is to maintain the discrepancy at the reference level X, the farther from X, the larger the correction magnitude. This naturally works for both directions where Y > X that decrease the discrepancy and where Y < X that increase the discrepancy. Half-ACRL has also matched the BF16 baseline, suggesting that ACRL’s adaptive correction magnitude remains effective, which results in a gap of only 0.08 percentage points with the full ACRL.
>
> (2)  In modern LLM post-training scenarios, avoiding excessively large training-inference discrepancy is more important than avoiding excessively small training-inference discrepancy, because a large train-inference gap can cause training collapse. We agree that, in practice, Y < X branch is less crucial than the Y > X branch, as being pointed our that the performance improvement is 0.08% (average), and 0.38 % (peak). We have updated the manuscript to clearly point out the order of importance in Remark 3.
>
> (3) In our ablation experiments, we established that the discrepancy should not be pushed excessively low($\alpha=\pm3$). If Half-ACRL with Y > X has already achieved desirable performance, the Y < X branch may be excluded. However, adding the Y < X branch is beneficial, because it directly eliminates the possibility that an excessively small discrepancy is left without correction action and it actively restores the discrepancy towards the reference value X, thereby improving robustness of ACRL. We have revised the manuscript by changing the expression of "necessary" into "beneficial" to avoid overstating the empirical evidence for the Y < X branch.

---

### Review · Reviewer_xgTr · 2026-06-23

**Summary Of Contributions:**

This paper aims to solve the LLMs' training-inference discrepancy problem.

The core contribution of this paper can be summarized as follows:
1. It proposes Adaptive Control Reinforcement Learning (ACRL) algorithm to adaptively control the discrepancy between training and inference within a reasonable range to stabilize the training.
2. Experiments show that ACRL enables stable RL training under aggressive quantization schemes, with the accuracy matching the BF16 baseline and outperforming baselines.

**Audience:**

Yes

**Audience Explanation:**

I think the TMLR audience might be interested in the findings of this paper, as it provides a systematic analysis of the training-inference discrepancy problem. Although the proposed method is more like a trivial engineering trick, it can be generally integrated into existing RL training frameworks to mitigate this problem.

**Broader Impact Concerns:**

I do not have concerns regarding the broader impact of this paper.

**Claims And Evidence:**

Yes

**Claims Explanation:**

1. Equation 5 defines the token-level discrepancy as $d(a_{i,t},\theta_{\mathrm{old}})=|\pi-\mu|$, and Equation 9 defines the adaptive exponent as $\alpha=\gamma\cdot \mathrm{sign}(A_{i,t})(1-Y/X)$. This formulation is reasonable, as it adaptively increases or decreases the correction strength depending on whether the current discrepancy is above or below the reference value.

2. On GSM8K, ACRL achieves an average accuracy of 87.05 which is slightly higher than the BF16 baseline at 86.60. On the 7B difficult math experiments, ACRL with $\gamma=0.65$ reports the best average accuracy, achieving 36.39% compared with 35.11% for BF16, although ACRL with $\gamma=0.35$ falls below BF16. The paper also has positive results for PPO/DAPO, Qwen3-32B, Qwen3-30B-A3B MoE, and MMLU-Pro. Overall, these results support the claim.

**Requested Changes:**

1. The authors use $s$ to denote the prompt, which is not standard in RL. In RL, $s$ typically denotes the state, which may only be partially determined by the prompt and may also depend on the broader environment or system context.
2. There is an ambiguity regarding the behavior policy used for low-precision inference and the old policy used in the GRPO importance ratio. Both seem to be parameterized by $\theta_{\mathrm{old}}$. Do $\mu$ and $\pi_{\mathrm{old}}$ share the same parameters? If so, why not use the same notation to represent them?
3. The GRPO objective should be written as learning or optimizing a policy, e.g., $J(\theta)$, rather than learning a prompt, e.g., $J(s)$.
4. I recommend replacing the term “quantization tax” in Section 5.1 with a plain explanation, since it is not a well-established term in the field (though understandable).
5. Equation 10 is too trivial to be presented as a standalone mathematical equation. I suggest moving it inline.

---

> ### Author Response · Authors · 2026-07-14
> **Author Response to Reviewer xgTr**
>
> Requested Changes:
>
> 1. We have revised our notation to align with standard conventions. Throughout the revised manuscript, we replaced the symbol s, previously used to denote the prompt, with q.
>
> 2. Thank you for pointing out this potential ambiguity. To clarify, both policies do share the exact same parameters $\theta_{old}$. However, we intentionally maintain separate notations to emphasize the physical separation of the engines. $\pi_{\theta_{old}}$ represents the distribution generated within the high-precision training environment (e.g., FSDP), while $\mu_{\theta_{old}}$ represents the distribution from the low-precision inference environment (e.g., vLLM). Because aggressive quantization and kernel differences cause these distributions to diverge, using a unified notation would inaccurately imply they output identical probabilities. We have updated the text in the Preliminaries section to explicitly state that they share $\theta_{old}$ while explaining the necessity of the distinct notations.
>
>
> 3.	We have fixed the notations in Equations 1, 2, 3, and 7. The GRPO objective has been written as learning a policy parameterized by $\theta$.
>
>
> 4. We have removed the phrase 'quantization tax' from Section 5.1 and replaced it with a plain description of the phenomenon (the accuracy degradation typically caused by low-precision quantization).
>
> 5.	We have now presented the Equation 10 inline.

---

### Review · Reviewer_bWQd · 2026-07-06

**Summary Of Contributions:**

The paper discusses the well established mismatch between training and inference in RL for LLMs. In particular, it tackles the FP8 inference vs. the common FP16 higher precision training. The authors propose ACRL as a method to keep the mismatch between the inference policy and the training policy around a particular reference level through adaptively reweighting token level gradients. The method computes an initial reference discrepancy, measures the current discrepancy, and reweights token level policy gradients using an adaptive exponent. ACRL reduces the update when the discrepancy is too large in the "wrong" direction and amplifies it when the discrepancy has become "too small". The strongest result is the discrepancy control evidence; FP8 training can collapse whereas ACRL maintains the measured discrepancy more stable. However, the papers empirical results do not convincingly support the claims about exploration nor exceeding BF16.

**Audience:**

Yes

**Audience Explanation:**

Efficient FP8 rollouts are a highly relevant task. The training inference mismatch is presented well. It would likely be interesting to TMLR readers in framing the inference mismatch.

**Broader Impact Concerns:**

No concerns.

**Claims And Evidence:**

No

**Claims Explanation:**

The stabilization claim is partly supported. The discrepancy plots demonstrate that uncorrected FP8 can diverge and that ACRL keeps the measured discrepancy controlled in the reported settings.

The claim that ACRL reliably improves accuracy, eliminates the quantization issues, and improves exploration are insufficient. The control argument assumes that the discrepancy can be written as $E=f(Q,W)+\eta$, drops $\eta$, and argues that forcing $E$ too small restricts the feasible weight space. This is not sufficiently rigorous to justify actively increasing the discrepancy when $Y<X$. Additionally, the ACRL weight is not clearly derived from the stated discrepancy metric; the discrepancy is $d=|\pi-\mu|$, however, the update uses a ratio based weight, $\rho = \left(\frac{\pi}{\mu}\right)^{\gamma\text{sign}(A)(1-Y/X)}$.

In the TIS baseline Eq. 2, the PPO/GRPO update ratio $\pi_\theta / \pi_{\theta_\text{old}}$ is used and not the rollout training correction ration.

Finally, the accuracy evidence lacks uncertainty. ACRL is not consistently best across tables and the reported improvements are relatively small.

**Requested Changes:**

The estimator notation should clearly distinguish the PPO/GRPO update ratio from the rollout training correction ratio. The current argument for avoiding too small discrepancy is unconvincing and needs support. The exploration claim needs to be revised to avoid equating $\pi > \mu$ with high absolute probability; add direct exploration metrics or comparisons to entropy regularization. Uncertainty should be reported with all of the empirical results.

---

> ### Author Response · Authors · 2026-07-14
> **Author Response to Reviewer bWQd Part 1**
>
> **Clarification on the ACRL Weight Derivation**:
>
> While we measure the system-level discrepancy using the absolute difference $d = |\pi - \mu|$ to evaluate the state of the system against our control thresholds ($X$ and $Y$), the mathematical correction must align with standard policy gradient theorems. Importance sampling corrections inherently require a multiplicative ratio to adjust the expected value of the gradient. Therefore, we deliberately decouple the measurement metric from the correction mechanism, applying the adaptive control as an exponent on the standard importance ratio $\pi/\mu$.

---

> ### Author Response · Authors · 2026-07-14
> **Author Response to Reviewer bWQd Part 2**
>
> Requested Changes:
>
> **1.** Thank you for catching this typo. We mistakenly defined the TIS correction ratio ($\rho_{i,t}$) in Equation 2 using the GRPO update ratio formula, omitting the inference policy $\mu$. We have corrected it in the revised manuscript as follows:
>
> $\rho_{i,t} = \frac{\textcolor{blue}{\pi}(a_{i,t}|s, a_{i,<t},\theta_{old})}{\textcolor{red}{\mu}(a_{i,t}|s, a_{i,<t},\theta_{old})}.$
>
> This now clearly separates the rollout training correction from the policy update ratio.
>
> **2.** ACRL is designed with robustness in mind. The core objective is to maintain the discrepancy at the reference level X, the farther from X, the larger the correction magnitude. This naturally works for both direction: when Y > X, it decreases the discrepancy, and when Y < X, it increases the discrepancy.
>
> We have updated the manuscript with the following:
>
> **a. Rigor of the Formulation:** We have revised section 3.1 using a constrained-optimization formulation to argue our claim more directly. The revised analysis shows that imposing an excessively small discrepancy bound reduces the feasible parameter space and may exclude optimal policies from being selected, therefore supporting our claim that the discrepancy cannot be excessively small.
>
> **b. Empirical Justification for the $Y < X$ Rule:** To provide concrete empirical proof, we conducted a targeted ablation study comparing the full ACRL framework against two alternative strategies when $Y < X$. The empirical results directly validate our theoretical formulation:
>
> *   **I. Continuous Discrepancy Reduction (Forcing to Zero):** In this aggressive variant, which explicitly pushes the discrepancy to zero, we completely removed the lower-bound protection and continuously penalized the discrepancy (using $\alpha = \pm 1$ and $\alpha = \pm 3$). With aggressive coefficients ($\alpha = \pm 3$), training directly and immediately collapses. With standard coefficients ($\alpha = \pm 1$), the model survives but suffers significant performance degradation (dropping -0.91% in peak accuracy compared to full ACRL).
> *   **II. Half-ACRL / Fallback to GRPO Variant:** In this milder variant (using $\gamma = 0.35$), when $Y < X$, we turn off ACRL and fall back to standard GRPO updates rather than adaptively controlling the discrepancy.It ultimately underperforms the full ACRL method (-0.38% vs. ACRL).
>
> **Continuous Reduction Variant:**
>
> | Method | Acc | Avg. Acc (last 200 steps) |
> |---|---|---|
> | ACRL Continuous Reduction | 87.19% | 86.14% |
> | vs. BF16 Baseline | -0.53% | -0.46% |
> | vs. Full ACRL | -0.91% | -0.91% |
>
> **Half-ACRL / Fallback to GRPO Variant:**
>
> | Method | Acc | Avg. Acc (last 200 steps) |
> |---|---|---|
> | FP8 Passive Optimization | 87.72% | 86.97% |
> | vs. BF16 Baseline | 0.00% | +0.37% |
> | vs. Full ACRL | -0.38% | -0.08% |
>
> In a standard BF16 pipeline with well-established RL infrastructure, training-inference errors are largely random. However, combining BF16 training with FP8 inference creates a fundamental systematic error. Blindly minimizing this discrepancy forces the high-precision training model to mimic lower-accuracy, low-precision inference results, actively degrading RL performance. Therefore, maintaining a controlled gap is necessary; to preserve model accuracy, the discrepancy must not be excessively reduced.

---

> ### Author Response · Authors · 2026-07-14
> **Author Response to Reviewer bWQd Part 3**
>
> Requested Changes
>
> **3.** We appreciate the reviewer pointing out this mathematical imprecision. We completely agree that equating the relative condition ($\pi > \mu$) with a strict absolute 'high probability' boundary was imprecise. As you correctly pointed out, Figure 6 contains instances where $\pi < \mu$ still holds a high absolute probability. However, as demonstrated by the probability density functions in Figure 6, the two regions exhibit clear stochastic dominance. The integral area of the density function is 1, that is, for any arbitrary absolute probability threshold $P_0 \in (0, 1)$, we have
> $P(\pi > P_0 \mid \pi > \mu) + P(\pi \le P_0 \mid \pi > \mu) = 1$ and $P(\pi > P_0 \mid \pi < \mu) + P(\pi \le P_0 \mid \pi < \mu) = 1$.
> According to Figure 6, the conditional probabilities satisfy:
>
> $P(\pi > P_0 \mid \pi > \mu) > P(\pi > P_0 \mid \pi < \mu), \forall P_0 \in (0, 1)$
>
> and
>
> $P(\pi < P_0 \mid \pi > \mu) < P(\pi < P_0 \mid \pi < \mu), \forall P_0 \in (0, 1).$
>
> This mathematically confirms that the distribution of tokens in the $\pi > \mu$ region is statistically heavily skewed toward higher absolute probabilities compared to the $\pi < \mu$ region. To resolve the ambiguity and ensure our definitions mathematically align with your observation, we have reframed Section 4.3 around these relative probability dynamics. We revised the terminology to 'Relative High-Probability' and 'Relative Low-Probability' to explicitly denote the relativity between the training policy $\pi$ and the reference policy $\mu$. This ensures our argument regarding how ACRL steers exploration relies strictly on the mathematical direction of the probability shift, avoiding any reliance on absolute probability thresholds.
>
> **4.**  Regarding your request for direct exploration metrics, we would like to respectfully point out that we have included comprehensive token entropy tracking over training steps in Figure 8 and Figure 10b. These continuous entropy plots serve as the direct exploration metrics requested, empirically demonstrating that ACRL maintains a wider, more exploratory distribution (higher entropy) than the uncorrected baselines.
>
> **5.** We completely agree that reporting standard deviations across multiple independent training runs is the gold standard for RL evaluation. However, due to the extreme computational overhead of training 3B, 7B, and 32B models, conducting multiple independent training runs for every configuration and baseline was prohibitively expensive for this study.
>
> To compensate for the inability to provide run-to-run variance, we implemented a rigorous temporal averaging methodology. As detailed in our results (e.g., Tables 1 and 5), we report the average pass@1 accuracy over the final 200 training steps during convergence. Averaging across these 200 distinct policy snapshots effectively smooths out step-to-step variance, providing a highly stable and robust indicator of the model's reasoning capability without the noise of a single-step anomaly.
>
> Furthermore, we designed our experiments to demonstrate robustness through broad generalization across highly diverse settings. We evaluated ACRL across multiple model scales (3B, 7B, 32B) and architectures (Dense, MoE), multiple RL algorithms (GRPO, PPO, DAPO), and diverse benchmark suites (Math and MMLU-Pro).
>
> Regarding the magnitude of the reported improvements, we acknowledge that the absolute numerical gains over the high-precision baseline may appear modest on certain isolated sub-metrics (such as AIME25). However, we emphasize that the primary objective of ACRL is optimization stability under extreme low-precision constraints.
>
> As demonstrated by the training curves in our evaluation, conventional importance sampling mechanisms suffer from severe instability and gradient spikes when subjected to FP8 inference discrepancies. ACRL fundamentally resolves this, delivering smooth, stable learning dynamics without the risk of training collapse.
>
> Therefore, the most critical measure of our method's success is the consistent elimination of the performance degradation caused by low-bit quantization. Across all independent experimental axes, ACRL systematically recovers the accuracy lost to aggressive FP8 quantization, successfully matching or slightly exceeding the high-precision BF16 baseline. The probability of a statistical anomaly producing this smooth, consistent recovery across such widely varying architectures and algorithms is vanishingly small.
>
> To provide full transparency regarding our experimental design and to emphasize the robustness of our evaluation strategy, we have added a 'Computational Constraints and Evaluation Robustness' subsection to our Experiments section.

---

### Review · Reviewer_bQ44 · 2026-07-11

**Summary Of Contributions:**

The paper tackles the training-inference mismatch in LLM RL post-training, specifically when using FP8 for rollouts and BF16 for training. They introduce ACRL, a method that dynamically reweights token-level gradients based on a sequence-level discrepancy metric to keep the mismatch near a reference baseline. The authors claim this not only stabilizes training but also inherently boosts exploration and eliminates the accuracy drop usually caused by quantization.

**Audience:**

Yes

**Audience Explanation:**

Despite my criticisms of the execution and analysis, the underlying problem is undeniably important. The training-inference mismatch caused by separate FP8 rollout engines and BF16 training backends is a massive headache for practitioners building modern RLHF pipelines. The community is actively looking for robust ways to handle low-precision rollouts without sacrificing alignment quality, so a solid solution to this problem would definitely be of interest to the TMLR audience.

**Claims And Evidence:**

No

**Claims Explanation:**

1. The core mathematical formulations contain fundamental errors that make it impossible to verify the actual implementation. Equation 2 incorrectly defines the importance sampling ratio using the PPO update ratio ($\pi/\pi_{old}$) instead of the actual rollout-training correction ratio ($\pi/\mu$). Furthermore, Equation 7 literally shows a subtraction operation for the gradient estimator instead of a multiplicative scaling.
2. The empirical evidence lacks statistical rigor. The reported accuracy gains are marginal (often <1% absolute) and are based on single-seed runs without error bars or confidence intervals. In the highly stochastic context of RL training, these minor differences could easily be attributed to run-to-run variance rather than the efficacy of ACRL.
3. The theoretical motivation for why "the discrepancy should not be excessively small" (Section 3.1) is conceptually flawed. The authors model the discrepancy as a differentiable function of weights and argue that forcing it to zero restricts the feasible weight space. This completely ignores the fact that the mismatch is primarily driven by discrete, non-differentiable system artifacts like FP8 hardware rounding and backend kernel inconsistencies.
4. The exploration analysis (Section 4.3) conflates quantization noise with semantic uncertainty. The authors claim that amplifying tokens where $\pi < \mu$ enhances exploration, but this deviation is largely driven by FP8 rounding errors rather than intrinsic semantic confidence. Moreover, simply showing an increase in training entropy (Figure 8) is insufficient; without generation-level diversity metrics, it remains unclear whether the model is exploring useful reasoning paths or merely degenerating into high-entropy garbage tokens.

**Requested Changes:**

1. Fix the mathematical typos in Eq. 2 and Eq. 7 immediately, and clearly distinguish the PPO update ratio from the IS correction ratio across all equations.
2. Disclose full training budgets for large-scale models. For the 32B and MoE experiments, provide exact step counts and compute budgets, and demonstrate whether the reported stability and accuracy gains actually persist over significantly longer training horizons.
3. Improve the visual presentation and formatting of the manuscript. (Many figures are blurry and some figures are disproportionately large)
4. Analyze hyperparameter safe zones and failure modes. Define the safe ranges for $\gamma$ and $C$ across different model scales. Discuss what happens when $\gamma$ is too large or $C$ is too tight/loose. Do you observe oscillatory dynamics or the controller over-amplifying spurious advantages?
5. Clarify the 0.1% overhead claim (Eq. 10). Explicitly state whether this includes the cross-device synchronization overhead between vLLM and FSDP, or if it only measures local GPU compute time.
6. Detail the quantization and probability extraction implementation. Explain exactly how $\mu$'s token probabilities are extracted from the FP8 inference engine (e.g., softmax temperature, normalization) and verify their numerical consistency with the training engine's $\pi$ to isolate pure quantization effects from other backend mismatches.

---

> ### Author Response · Authors · 2026-07-14
> **Author Response to Reviewer bQ44 Part 1**
>
> Accurate, convincing and clear evidence concerns:
>
> **1.** Answered in Requested Changes 1 below.
>
> **2.** While we acknowledge that running multiple independent seeds is a historical standard for small-scale RL, conducting multi-seed evaluations for 3B, 7B, and 32B large language models across various baseline configurations (BF16, TIS, MIS and ACRL) requires a prohibitive compute budget and is not standard practice in modern LLM post-training research.
>
> To ensure rigorous statistical confidence without relying on multi-seed variance, our evaluation protocol leverages two robust stabilization techniques:
>
> **a. Temporal Averaging:** Rather than reporting a peak single-step anomaly, we report the average pass@1 accuracy over the final 200 training steps during convergence (as detailed in Tables 1 and 5). Averaging across 200 distinct policy snapshots completely smooths out step-to-step stochasticity, providing a highly stable indicator of true reasoning capability.
>
> **b. Broad Generalization:** We evaluated ACRL across multiple model scales (3B, 7B, 32B), divergent architectures (Dense, MoE), multiple RL algorithms (GRPO, PPO, DAPO), and diverse benchmark suites (Math and MMLU-Pro). The probability of a single "lucky" statistical anomaly producing consistent recovery across all of these independent experimental axes simultaneously is vanishingly small.
>
> Regarding the magnitude of the reported improvements, we respectfully disagree that the gains should be measured purely as absolute accuracy leaps over the high-precision baseline. The primary objective of ACRL is optimization stability and baseline (BF16) recovery under low-precision constraints.
>
> As our training curves demonstrate, conventional importance sampling mechanisms suffer from severe gradient spikes and training collapse under FP8 inference. ACRL systematically eliminates this degradation. The most critical measure of our method's success is not achieving massive accuracy gains over BF16, but rather successfully matching the BF16 baseline while operating within the highly constrained, unstable FP8 environment. Because ACRL consistently delivers smooth learning dynamics compared to previous importance sampling fixes and recovers to the BF16 baseline in all experiments, the empirical evidence firmly supports its efficacy in enabling stable low-precision inference for LLM post-training pipelines.
>
> We have added a brief 'Evaluation Robustness' subsection to our Experiments section, which discusses these specific design choices and clarifies the overall robustness of our experimental framework.

---

> ### Author Response · Authors · 2026-07-14
> **Author Response to Reviewer bQ44 Part 2**
>
> Accurate, convincing and evidence concerns:
>
> **3.** We agree with the reviewer that FP8 rounding and kernel mismatches are discrete hardware artifacts.
>
> However, the formulation in Section 3.1 does not attempt to calculate a gradient through the hardware rounding itself. To be explicitly clear, we do not take derivatives of the error equation $E = f(Q, W) + \eta$. This equation serves strictly to model the composition of discrepancy and the noise. During optimization, we simply use $\mu$ and $\pi$ to compute the ratio $\rho$ to perform standard policy gradient updates.
>
> Because the FP8 inference probabilities ($\mu$) contain these hardware errors, forcing our training model to match them exactly (a discrepancy of zero) is harmful. It forces the high-precision training model to memorize random hardware glitches rather than learning the actual mathematical reasoning task.
>
> This is exactly why ACRL prevents the discrepancy from being excessively small. It acts as a necessary buffer, giving the model enough flexibility to ignore the hardware noise.
>
> **4.** First, we acknowledge that quantization errors naturally contribute to the discrepancy between $\pi$ and $\mu$; we actively repurpose this quantization-induced discrepancy for guided exploration. Instead of a blanket amplification when $\pi < \mu$, ACRL applies a fine-tuned approach: Quadrants I and III amplify tokens that promote exploration, whereas Quadrants II and IV actively penalize those that discourage it. Ultimately, AGPA harnesses this raw FP8 quantization noise and actively steers it into conditional and directional exploration paths.
>
> In response to the concern that exploration is largely driven by FP8 rounding errors, we note that in Figure 8, the FP8 baseline exhibits the lowest entropy before it runs into training collapse. Thus, the mere introduction of FP8 rounding errors does not boost entropy or enhance exploration; entropy only peaks into unbounded regions once training has already collapsed. By sustaining a higher level of meaningful entropy than all the FP8, TIS and MIS baselines without collapsing, this experiment proves the efficacy of AGPA in actively enhancing exploration.
>
> On top of showing the increased entropy in Figure 8, ACRL achieves superior accuracy to further consolidate that the exploration is genuine and meaningful. ACRL systematically boosts mathematical reasoning accuracy across various strict, exact-match benchmarks (GSM8K, Difficult Math, MMLU-Pro) and algorithms (PPO, GRPO, DAPO). It is impossible to achieve this peak accuracy by generating high-entropy "garbage" tokens, as seen in the case of the abnormally high entropy observed in the MIS baseline during the 7B Difficult Math test, which delivers inferior test accuracy.

---

> ### Author Response · Authors · 2026-07-14
> **Author Response to Reviewer bQ44 Part 3**
>
> Requested Changes:
>
> **1.** Thank you for catching the typo in Equation 2. We mistakenly defined the TIS correction ratio ($\rho_{i,t}$) using the GRPO update ratio formula, omitting the inference policy $\mu$. We have corrected it in the revised manuscript to clearly separate the rollout training correction from the policy update ratio:
>
> $\rho_{i,t} = \frac{\textcolor{blue}{\pi}(y_{i,t}\vert{}x, y_{i,<t},\theta_{old})}{\textcolor{red}{\mu}(y_{i,t}\vert{}x, y_{i,<t},\theta_{old})}$
>
> Regarding Equation 7, we respectfully clarify that the operation is indeed a multiplicative scaling, not a subtraction. The symbol situated at the beginning of the second line is a standard mathematical multiplication dot ($\cdot$). We recognize that PDF rendering engines or automated text-parsing tools can misread a line-broken dot as a hyphen or minus sign, but the mathematical notation correctly reflects the expected multiplicative operation. We will retain the current formulation for Equation 7.
>
> **2.** We agree that transparency regarding computational budgets is critical for reproducibility in large-scale RL systems. We have updated the appendix to explicitly outline the step counts and compute infrastructure for our large-scale experiments.
>
> **a. Compute Budget and Step Counts:** The models were trained utilizing 4 Nvidia B200 GPUs (180GB memory per GPU). Both the Qwen3-32B (Dense) and Qwen3-30B-A3B (MoE) were trained for a total of 800 steps, requiring 21 hours and 46.5 hours (1.94 days) respectively. We note that the MoE model's extended duration was due to a conservative inference concurrency configuration utilized during that specific run.
>
> **b. Training Horizons and Post-Convergence Stability:** The reviewer asked whether stability persists over significantly longer training horizons. We would like to highlight that both models actually reach convergence well before the end of the training run, stabilizing around step 400. At step 400, the Qwen3-32B (Dense) already achieves an accuracy of 95.51% using ACRL (closely matching the 95.53% BF16 baseline), and the Qwen3-30B-A3B (MoE) achieves 96.21% using ACRL (outperforming the 95.22% BF16 baseline). Therefore, our evaluation extending to 800 steps was deliberately designed to test this exact post-convergence scenario. While uncorrected FP8 pipelines typically suffer from discrepancy explosion and training collapse early in the run, our results demonstrate that ACRL not only reaches convergence but remains perfectly stable for hundreds of steps afterward, maintaining peak evaluation accuracy without oscillatory degradation or collapse.
>
> **3.** Thank you for the feedback. We have re-rendered all figures at a higher resolution to eliminate any blurriness and adjusted their scaling to ensure appropriate proportions throughout the revised manuscript.
>
> **4.** We appreciate the reviewer's detailed questions regarding hyperparameter sensitivity. We have already documented the primary safe zones and failure modes for $\gamma$ in Section 5.4 (Table 9), demonstrating that medium-level control ($\gamma \in [0.5, 1.7]$) yields optimal stability across tested scales, while extreme values lead to over-constraint ($\gamma \ge 2.1$) or instability ($\gamma = 0.1$).
>
> To directly answer your questions regarding the clipping parameter $C$:
>
> **a. Oscillatory Dynamics:** The primary purpose of $C$ is exactly to prevent the oscillatory dynamics and over-amplification of spurious advantages you mentioned. If $C$ is unbounded or too loose, token-level outliers can violently over-amplify the gradient and destabilize training. Conversely, if $C$ is too tight, the necessary corrective magnitude is neutered.
>
> **b. Safe Ranges:** Empirically, a tight bound of $C = 3$ safely prevents oscillation for standard FP8 quantization across our tested scales, while extreme environmental distribution shifts (e.g., Section 5.3) require loosening it to $C = 5$.
>
> Because $C$ is inherited directly from established TIS and MIS baseline conventions, we believe that expanding the manuscript to include low-level sensitivity analysis would distract from the core theoretical focus of the paper. Therefore, to keep the text concise and centered on resolving the fundamental training-inference mismatch, we provide this specific clarification here in the rebuttal rather than appending a new sensitivity analysis to the main text.

---

> ### Author Response · Authors · 2026-07-14
> **Author Response to Reviewer bQ44 Part 4**
>
> Requested Changes:
>
> **5.** To directly address your question regarding the overhead calculation: the 0.059-second ACRL overhead measures the isolated local compute time required to calculate the importance sampling weight tensor.
>
> Regarding the network transmission: because the training engine holds the policy weights ($\theta$), the inference probabilities ($\mu$) must indeed be transmitted back to the trainer via the collective communication backend (e.g., HCCL or NCCL). However, the additional cross-device payload introduced by this step is mathematically negligible. In standard RL pipelines, the rollout engine is already required to transmit the generated token IDs and environment rewards back to the trainer. Appending $\mu$ to this existing network transfer adds only a single scalar per generated token.
>
> To quantify this overhead precisely using the exact profiling environment (Qwen2.5-3B configuration, Appendix Table 10): with a training batch size of 64 prompts, 8 rollout responses per prompt, and a maximum response length of 2048 tokens, the theoretical maximum generated tokens per batch is 1,048,576. Transmitting the FP32 $\mu$ scalars for all generated tokens results in a total additional payload of 4 MB per training step.
>
> On standard AI cluster interconnects operating at 400 Gbps (50 GB/s), the theoretical transmission time for this 4 MB payload is approximately 0.08 milliseconds ($0.00008$ seconds). Because this network latency is entirely absorbed by the baseline cross-node synchronization, the 0.059 seconds of measured local GPU compute accurately reflects the only measurable bottleneck introduced by ACRL, fully validating the 0.1% overhead claim. We provide this payload analysis here to address your question directly without expanding the manuscript's methodology section.
>
> **6.** To directly address your question regarding the extraction implementation: there is absolutely no custom quantization or probability extraction code required. Returning the generated token probabilities ($\mu$) is not a specialized feature; it is a fundamental, native API output present in virtually every modern LLM inference engine (including vLLM, TensorRT-LLM, and standard Hugging Face pipelines).
>
> Because we simply utilize this universally available, native API output (returning standard unscaled log-probabilities), no manual softmax temperature adjustments or custom mathematical normalizations are applied on our end. This inherently guarantees numerical consistency with the engine's exact internal state.
>
> Furthermore, attempting to artificially isolate "pure quantization effects" from other backend mismatches is unnecessary for our method. As defined in Section 3.1, ACRL operates directly on these natively returned probabilities to adaptively correct the total system-level discrepancy. This naturally and intentionally captures both the FP8 precision loss and any underlying backend engine inconsistencies simultaneously.
>
> Because retrieving $\mu$ requires only a trivial, standard API call rather than a novel algorithmic extraction method, we provide this structural clarification here to answer your question directly without appending standard software documentation to the main manuscript.